# Functional annotation of proteins for signaling network inference in non-model species

Lisa Van den Broeck [1] ✉, Dinesh Kiran Bhosale[2], Kuncheng Song[3], Cássio Flavio Fonseca de Lima [4,5], Michael Ashley[2], Tingting Zhu [4,5], Shanshuo Zhu[4,5], Brigitte Van De Cotte[4,5], Pia Neyt[4,5], Anna C. Ortiz[6], Tiffany R. Sikes[6], Jonas Aper[7], Peter Lootens [8], Anna M. Locke[6,9], Ive De Smet [4,5] & Rosangela Sozzani [1] ✉

Molecular biology aims to understand cellular responses and regulatory dynamics in complex biological systems. However, these studies remain challenging in non-model species due to poor functional annotation of regulatory proteins. To overcome this limitation, we develop a multi-layer neural network that determines protein functionality directly from the protein sequence. We annotate kinases and phosphatases in *Glycine max*. We use the functional annotations from our neural network, Bayesian inference principles, and high resolution phosphoproteomics to infer phosphorylation signaling cascades in soybean exposed to cold, and identify Glyma.10G173000 (TOI5) and Glyma.19G007300 (TOT3) as key temperature regulators. Importantly, the signaling cascade inference does not rely upon known kinase motifs or interaction data, enabling de novo identification of kinase-substrate interactions. Conclusively, our neural network shows generalization and scalability, as such we extend our predictions to *Oryza sativa*, *Zea mays*, *Sorghum bicolor*, and *Triticum aestivum*. Taken together, we develop a signaling inference approach for non-model species leveraging our predicted kinases and phosphatases.

Molecular biology aims to understand the molecular base of cellular behavior, and growth and development of organisms as a result of interconnected biochemical pathways, gene regulation, and cell-to-cell interactions. Studying these interconnected networks and pathways in non-model organisms is critical, as model systems are not representative of their entire class[1]. Moreover, the inadequate functional annotation of regulatory proteins, including kinases or phosphatases that form the basis of signaling networks, hinders omics research in non-model species. Even within the model species *Arabidopsis thaliana*, 1229 (~4.5%) protein-coding loci remain unannotated. Functionally annotating proteins by classifying them into families provides clues to their structure, localization, and activity. Currently, the

[1]Plant and Microbial Biology Department and NC Plant Sciences Initiative, North Carolina State University, Raleigh, NC 27695, USA. [2]Electrical and Computer Engineering Department, North Carolina State University, Raleigh, NC 27695, USA. [3]Bioinformatics Research Center, North Carolina State University, Raleigh, NC 27695, USA. [4]Department of Plant Biotechnology and Bioinformatics, Ghent University, B-9052 Ghent, Belgium. [5]VIB Center for Plant Systems Biology, B-9052 Ghent, Belgium. [6]USDA-ARS Soybean & Nitrogen Fixation Research Unit, Raleigh, NC 27607, Belgium. [7]Protealis NV, Technologiepark-Zwijnaarde 94, 9052 Ghent, Belgium. [8]Plant Sciences Unit, Flanders Research Institute for Agriculture, Fisheries and Food (ILVO), 9090 Melle, Belgium. [9]Department of Crop and Soil Sciences and NC Plant Sciences Initiative, North Carolina State University, Raleigh, NC 27695, USA. ✉e-mail: lfvanden@ncsu.edu; rsozzan@ncsu.edu

dominant approach for functional prediction of protein families relies on hidden Markov models (HMMs) that identify significant protein sequence similarities or on alignments across a large database of annotated sequences[2,3]. These approaches have been successful but rely on conserved domains deduced from sequence alignments and are unable to identify higher-order correlations or structurally and evolutionarily diverse groups, such as phosphatases[4]. In the last decade, deep learning, a subdivision of machine-learning methods, has been deployed in biological sciences for various applications, ranging from classification questions to identifying hidden features in omics-datasets[5–7]. Thus, deep-learning methods provide an opportunity to circumvent sequence alignments and directly predict protein function from a sequence database. Several deep-learning algorithms have been developed to predict the function of DNA sequences. For example, DeepBind, DeepSea, DanQ, and TBiNet, each learning from past models, predict DNA sequence functions and TF-DNA binding sites[8–11]. Generally, within such neural networks, convolutional neural networks (CNNs) are deployed to scan for small stretches of conserved nucleotides to reflect spatial information of an input sequence. Although many studies have investigated the functionality of DNA sequences, relatively few studies attempt to predict the functionality of protein sequences. Protein sequences are inherently more complex as they consist of 20 possible notations compared to only four for DNA. Existing machine-learning approaches that predict protein function do so by predicting gene ontology or protein structure associated with the amino acid sequence[12–14]. For example, ProteinBERT is a deep-learning language model that inputs GO annotations and protein sequences and shows state-of-the-art performance on predicting protein structure, post-translational modifications, and biophysical properties[15]. A recent study showed the potential of machine-learning models to complement existing approaches for protein function prediction tools[16]. Specifically, ProtCNN, a neural network trained on protein domain sequences, accurately annotated protein domains and improved and expanded on current proteins annotations[16].

Combined, the functional annotation of kinase and phosphatase regulatory proteins and high-throughput-omics, like phosphoproteomics, allow for the inference of signaling networks. Specifically, the prior knowledge of regulatory proteins significantly improves network inference accuracy. Without prior knowledge of the upstream regulators, well-established network inference methods yield predictions that were not better than random guessing[17]. Network inference enables the prediction of undescribed signaling pathways forming hypotheses and guiding experiments[18]. Several analytical tools, such as GENIE3[17] and ARACNE[19], infer causal regulations among differentially expressed genes to, for example, identify key transcriptional regulators[20]. GENIST, a dynamic Bayesian network approach, leverages time series data to identify dependencies among the modeled genes[20–23]. However, such network inference approaches to identify dependencies among phosphorylated proteins is still a bottleneck. Current approaches generally create phosphorylation signaling networks using predicted protein substrates of kinases based on experimentally identified consensus sequence motifs or known protein-protein interactions supplemented with contextual information, such as subcellular compartmentalization, colocalization, and coexpression[24–26]. However, these approaches have limitations as for many kinases and phosphatases consensus motif sequences have not yet been comprehensively determined resulting in incomplete networks and contextual information might be missing. It is proposed that "data-driven" approaches that utilize phosphorylation intensities to directly infer relationships by identifying dependencies among phosphorylated peptides can aid in a network-level understanding of kinase activity. Moreover, to computationally construct accurate signaling networks, a priori knowledge on protein functions, such as kinase or phosphatase activity, is needed.

In this study, to generate a priori knowledge on regulatory proteins, including kinase and phosphatase activity, we developed a scalable approach that determines sequence functionality using deep learning. Specifically, we designed a multi-layer neural network to directly extract hidden features from any protein sequence and classify them into protein families. To show the generalization and scalability of our neural network, we annotated the proteome of six species, including *Arabidopsis thaliana*, soybean, wheat, maize, rice, and sorghum and focused on the kinase and phosphatase predictions. Lastly, we used our compiled kinase/phosphatase list to infer phosphorylation cascades in soybean, demonstrating that inference of signaling networks in non-model species can be facilitated through functional annotation. This approach allowed us to identify two key temperature response regulators in soybean, Glyma.10G173000 (TOI5) and Glyma.19G007300 (TOT3), and their 116 and 60 putative substrates, respectively. Overall, our results show that deep-learning models in combination with existing methods, such as HMMs, strongly improve protein function annotation, which can then significantly advance systems biology studies.

## Results

### Designing a neural network for functional annotation of non-model species

To understand cellular behavior and organismal growth and development as a result of interconnected pathways, we aim to unravel the interactions between genes, proteins, etc., within those systems. To computationally model interactions, a priori knowledge on protein functions, such as kinase or phosphatase activity, is needed. The conventional method to annotate protein function relies on hidden Markov models (HMMs), which classifies proteins into their respective family. Thus, to identify the undescribed phosphatases in soybean, we performed a HMMER search[3] with HMMs for Ser/Thr phosphatases (STs), dual-specificity phosphatases (DSPs), protein phosphatases 2C (PP2Cs), and protein tyrosine phosphatases (PTPs). We identified a total of 306 phosphatases, of which 109 (36%), 34 (11%), 160 (52%), and 3 (1%) belong to STs, DSPs, PP2Cs, and PTPs, respectively (Supplementary Data 1). To support the identification of the putative soybean phosphatases, we identified soybean orthologs from the previously described phosphatases of the model species *Arabidopsis thaliana*[27]. We identified 282 soybean orthologs, of which 224 (79.4%) were in common with the HMM-identified phosphatases. As such, 58 (20.6%) of the orthologs were missed. Because HMMs assume that each amino acid at a particular position is independent of the amino acids at all other positions, we reasoned that HMMER might miss some phosphatases since they represent a structurally and evolutionarily diverse protein family. In addition, HMMs might not detect all phosphatases as they cannot capture any higher-order (i.e., nonlinear) correlations. To overcome these limitations, we designed a neural network and included a recurrent layer that identifies dependencies between the amino acids at all positions, and activation functions that capture higher-order correlations. Our neural network, hereafter referred to as PF-NET (Protein Family classification NETwork), takes into account the entire sequence context to classify protein sequences into one of 996 protein families. The neural network architecture consists of four different layers: (1) a convolutional neural network layer (CNN) that extracts putative protein domains, i.e., the functional units of a protein, by performing a convolution across the sequences with a kernel size of 7 (Fig. 1a, see "Methods"), (2) an attention layer that emphasizes the learned patterns from the CNN layer by assigning increased importance to key domains, (3) a bidirectional long short-term memory (biLSTM) layer that captures long-distance dependencies within the sequences and between detected domains, and (4) two dense layers connected to the output vector (Fig. 1a). Therefore, this neural network is uniquely designed to address the current classification gaps.

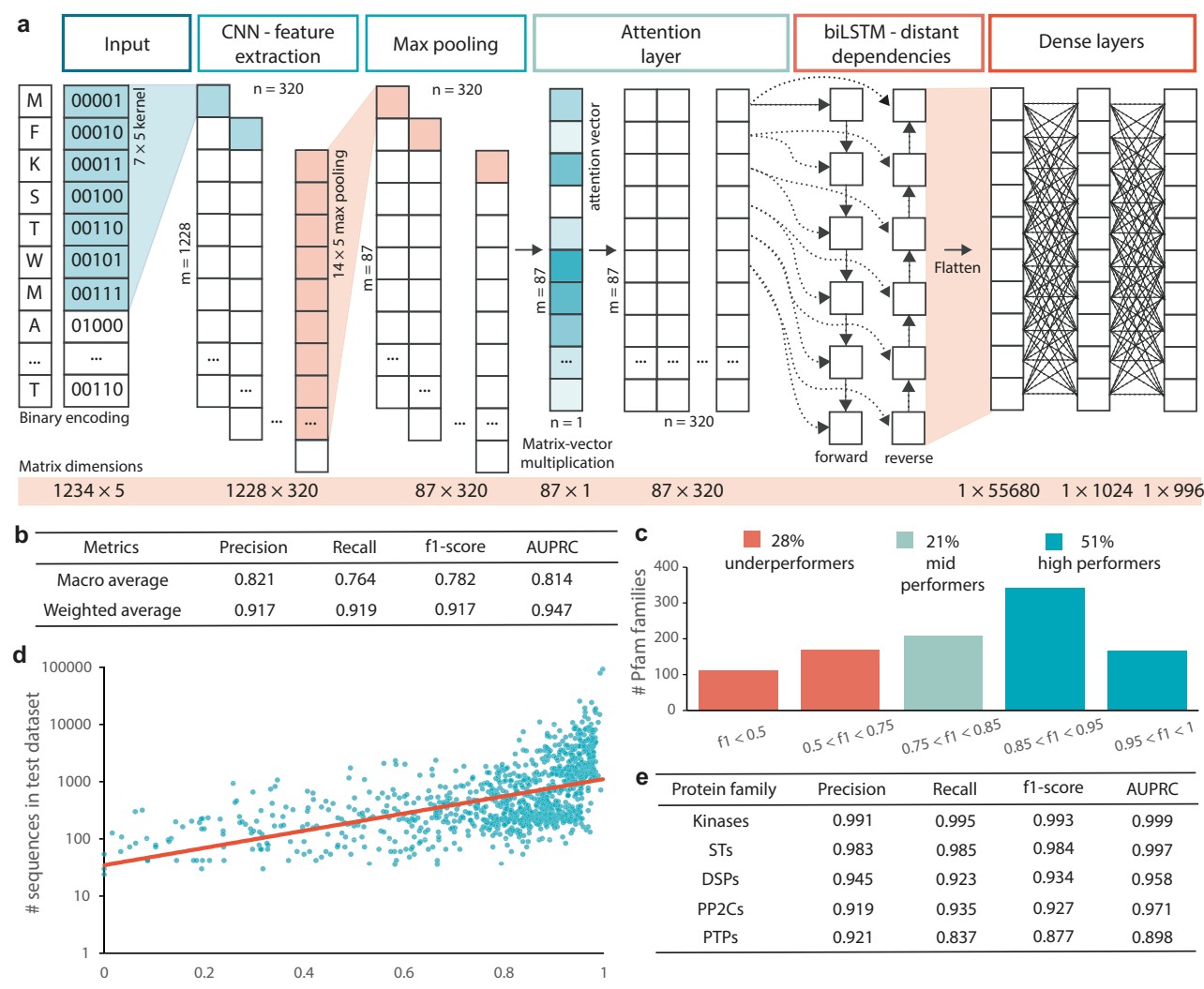

Fig. 1 | Classification performance of PF-NET (Protein Family classification NETwork). a Input amino acid sequences are binary encoded into 1234 × 5 matrices where each row is the position in the sequence and each column represents an amino acid. The 1D convolutional neural network (CNN) performs a convolution across the encoded sequences with a kernel size of 7, detecting any motifs. Element-wise multiplication is applied with the output of the CNN after max pooling and an attention vector. The output matrix is passed through a bidirectional long short-term memory (biLSTM) layer to identify any distant dependencies within the entire sequence. Two dense layers fully connect to the final 996-dimensional output vector. b The overall performance of PF-NET in terms of precision, recall, f1 score, and AUPRC (area under the precision-recall curve). For the macro average, these scoring metrics of each class were averaged. For the weighted average, the scoring metrics for each class were weighted by the number of true instances and then averaged to account for class imbalance. c The number of protein families within each of five f1-score intervals. 28%, 21%, and 51% are classified as underperformers, mid-performers, and high performers, respectively, and have an f1-score smaller than 0.75, between 0.75 and 0.85, and between 0.85 and 1, respectively. d A scatterplot of the number of sequences in the test dataset for the 996 protein families (represented exponentially on the x axis) and the corresponding f1 score. An exponential trend line colored in red was fitted onto the scatterplot. e PF-NET's performance in terms of precision, recall, f1 score, and AUPRC for the kinase and phosphatase proteins.

We then used this architecture to functionally annotate kinases and phosphatases. To this end, we trained and optimized PF-NET's hyperparameters, such as learning rate, number of filters, and activation function, to give the best performance (see "Methods"). To learn sequence features, such as structural disorder and transmembrane helices, PF-NET achieves a high overall accuracy of 91.9% and a weighted precision, recall, and f1 score of 91.7%, 91.9%, and 91.7%, respectively (Fig. 1b and Supplementary Fig. 1). To gain a better understanding of the contribution of the individual layers, we removed each layer and evaluated the performance of these models. This ablation study showed that especially the CNN and biLSTM are crucial for PF-NET's performance (Supplementary Fig. 2). We examined the performance of PF-NET in more detail for each of the 996 protein families (see "Methods") by evaluating these scoring metrics for the test dataset. In total, 50.9% of the protein families were classified as

high performers and had an f1 score above 85.0% (Fig. 1c, d and Supplementary Data 2). Generally, the high performers were represented by more sequences within the training dataset contributing to their high prediction scoring metrics (Fig. 1d). Among those were kinases and phosphatases (e.g., STs, DSPs, PP2C, and PTPs), suggesting that our neural network could outperform HMMs in identifying kinases and phosphatases (Fig. 1e). Overall, our regulatory proteins of interest showed a high f1 score between 87.7% and 99.3%, suggesting that our approach can advance the predictions of regulatory proteins.

To enable comprehensive studies of signaling networks, we aim to use PF-NET to predict the needed a priori knowledge on the regulatory proteins in soybean. However, the limited experimental validation of these regulatory proteins (kinases and phosphatases) in plants is a bottleneck to assess PF-NET's performance. Thus, to assess the performance of our neural network in functionally predicting kinases and

phosphatases, we used *Saccharomyces cerevisiae* (yeast) as a benchmark species. Specifically, we used the complete yeast proteome as an independent benchmark dataset, made functional predictions for all proteins, and focused on the phosphatases and kinases. To evaluate the predictions, we compared our neural network with the ground truth, here represented by a manually curated list of 115 and 38 biochemical experimentally validated kinases and phosphatases, respectively (Supplementary Data 3). Predictions in common with the ground truth were considered true positives. Predictions for which no experimental data were available were considered to be unconfirmed. Overall, we observed a high number of identified proteins. Notably, 34 of the 137 predicted kinases were classified as unconfirmed when compared to the ground truth. Because PF-NET outputs a probability distribution, where the probability for all 996 classes sums up to one for each input sequence, every sequence is categorized even if it does not belong to one of the 996 classes. We leveraged this output to assess the probability and prediction reliability associated with each prediction. For each sequence, we extracted the highest probable predictions (Fig. 2a, b). Almost all true positive predictions showed a probability of more than 95%, while the majority of unconfirmed predictions had a probability below 60% (Fig. 2a, b). To exclude low probable predictions in an unbiased fashion, we calculated the cost with each threshold using the true positive rate and the false discovery rate and selected the threshold with the lowest cost (Supplementary Fig. 3). After applying the selected threshold, we achieved a sensitivity of 89% and 66% for kinases and phosphatases, respectively (Fig. 2c–e). Using this approach, we unaccounted for only two proteins, the kinase RIO1 and the phosphatase DET1, that were experimentally validated, but had a prediction probability below the set threshold (Fig. 2a, b). We found similar performance metrics with HMMER, but some identified proteins were predicted to be kinases/phosphatases by PF-NET and not by HMMER (Fig. 2c, d). For example, the predicted kinase by PF-NET, CEX1, which was missed with HMMER, was shown to contain a kinase-like domain based on the crystal structure[28]. We also compared PF-NET's performance to ProtCNN, a protein classification neural network. However, as ProtCNN is trained on protein domain

sequences rather than the full protein sequence, a poor performance was retrieved (Supplementary Fig. 4). Overall, these results indicate that, with the high predictive power of our neural network, we can augment protein families.

## Identifying functional kinases and phosphatases in plants

Provided that our neural network generalized well (i.e., the ability of the neural network to digest independent data) for yeast proteins, we reasoned that our neural network would also generalize well toward plant proteins. To test this, we compared PF-NET's results to published computational protein classification studies and HMMER. Specifically, we functionally annotated the *A. thaliana* and soybean proteome, and selected kinases and phosphatases in *A. thaliana* and kinases in soybean[27,29,30] (Supplementary Data 4 and 5). Leveraging our advance from yeast, we applied a threshold for each protein family and compared our predictions with the published annotation studies to compute classification metrics (Supplementary Fig. 3). The neural network showed a recall of 97% and 95% for the *A. thaliana* kinases and phosphatases, respectively (Fig. 3a). Comparable metric performances were found with HMMER searches; however, a different set of classified proteins were identified with these two methods. Specifically, six annotated phosphatases, including the biochemically validated proteins AT1G05000[31] and PEN2 (AT3G19420)[32], were predicted by PF-NET, but missed with HMMER. Similarly, three predicted kinases (AT5G11360, AT4G29654, and AT1G61475) were undetected by HMMER, but showed a predicted INTERPRO kinase or kinase-like domain, suggesting a kinase role. We also observed a similar performance for the soybean kinases with a recall of 98% and a false discovery rate of 1% (Fig. 3b). Within the pool of newly predicted soybean kinases by PF-NET (20 proteins), we looked at their GO associations and found that 13 proteins have a molecular function or Panther description related to protein phosphorylation (Supplementary Data 6)[33–35]. These 20 proteins are potentially undescribed kinases, which were missed by HMMER, emphasizing the need for machine-learning annotation models. Overall, we demonstrated that PF-NET generalizes well toward truly independent plant datasets and can predict the functionality of proteins across species.

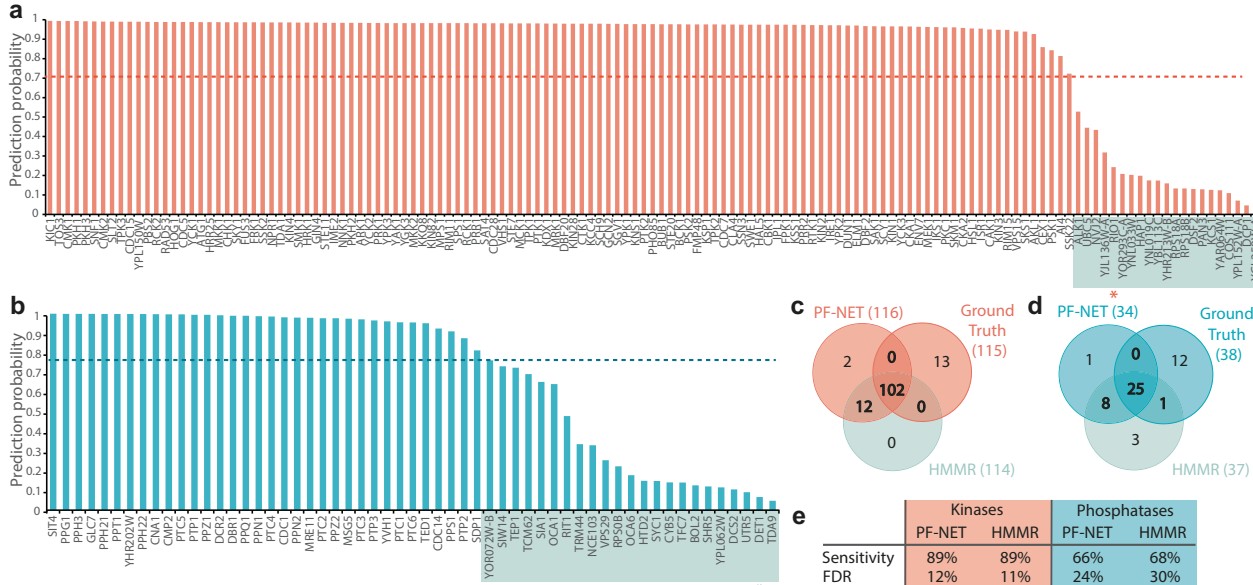

**Fig. 2 | Functional predictions of kinases and phosphatases in *Saccharomyces cerevisiae*. a**, **b** The prediction probabilities of all predicted kinases (**a**) and phosphatases (**b**) in yeast. The dotted line indicates a set threshold that maximizes the true positive rate. Predictions marked with a gray box fall below the threshold. Predictions that have a probability below the threshold marked with a star are

experimentally validated kinases or phosphatases. **c**, **d** Commonly identified kinases (**c**) and phosphatases (**d**) by PF-NET (Protein Family classification NETwork), HMMER search, and the ground truth. **e** The overall performance of PF-NET and HMMER in terms of sensitivity and false discovery rate (FDR).

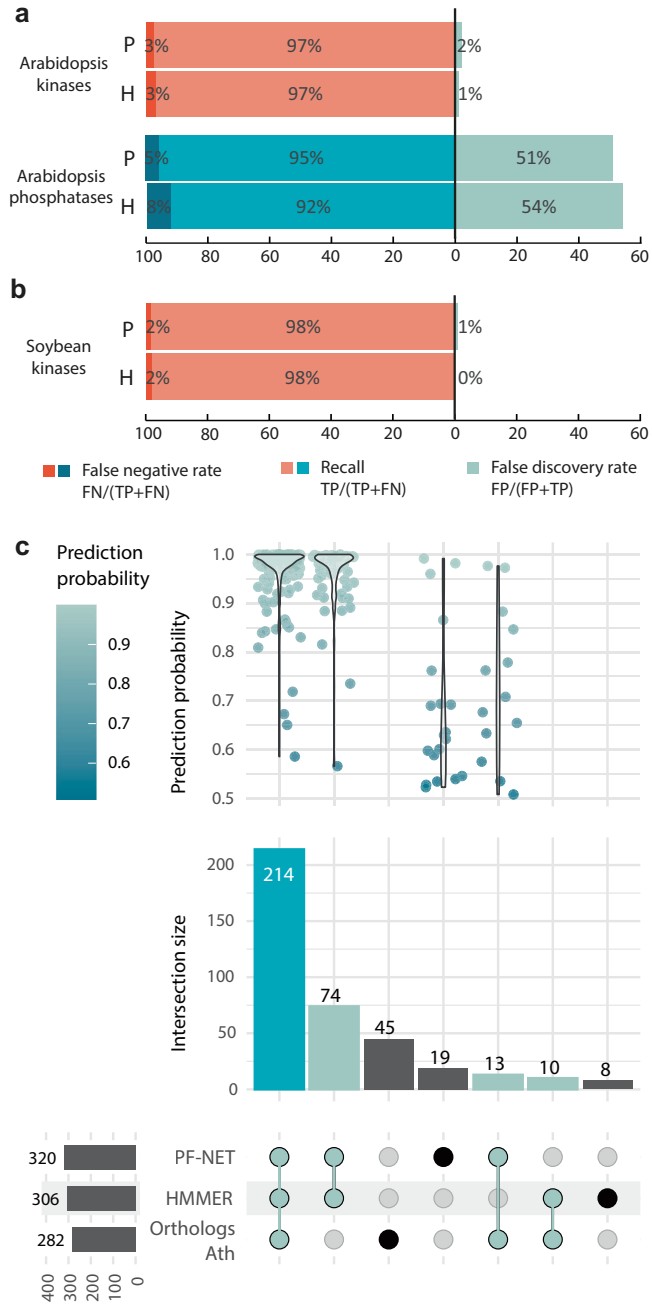

**Fig. 3 | Neural network predictions for the *A. thaliana* and soybean kinases and phosphatases. a, b** Percentage of false-negative rate, recall, and false discovery rate of PF-NET (P) and HMMER (H) compared to annotation research studies for **a** *A. thaliana* kinases[29], **a** *A. thaliana* phosphatases[27], and **b** soybean kinases[30]. **c** Commonly identified phosphatases by PF-NET, HMMER search, and ortholog search of *A. thaliana* (Ath) phosphatases in *Glycine max* (soybean). The top graph plots the prediction probabilities of the PF-NET predictions for each intersection, while the bottom bar graph plots the counts of each intersection. The horizontal bar graph on the bottom left shows the total predicted phosphatase proteins by each method. TN true negatives, FN false negatives, TP true positives, FP false positives.

Given the high generalization and prediction confidence of PF-NET, we explored the unpredicted soybean phosphatases. A well-annotated and comprehensive list of regulatory proteins in soybean would ensure the next steps, i.e., the inference of signaling networks. To this end, we used PF-NET and identified 320 soybean phosphatases

as compared to 306 and 282 identified with HMMER and *A. thaliana* orthologs, respectively (Fig. 3c and Supplementary Data 1). Of these, 214 were in common between all the annotation methods and showed high prediction probabilities from PF-NET. Importantly, an additional 13 *A. thaliana* orthologs were classified as phosphatases by PF-NET with variable prediction probabilities (Fig. 3c). Nineteen proteins were functionally annotated to phosphatases solely by PF-NET, of which three proteins had a probability above 95% (Fig. 3c). Notably, we found that two have a molecular function GO annotation of "protein serine/threonine phosphatase activity". Of the 311 phosphatases identified by at least two of the three methods (colored light green in Fig. 3c), 108 (35%), 41 (13%), 146 (47%), and 6 (2%) belong to STs, DSPs, PP2Cs, and PTPs, respectively. With this in mind, we combined the extended kinase list and the annotated phosphatases (see "Methods") to use as prior knowledge for the inference of phosphorylation cascades in soybean.

## Phosphorylation signaling network inference in soybean

Phosphorylation cascades are among the first steps in signal perception and are responsible for transducing environmental and cellular signals. To assess how signaling pathways dynamically rewire upon environmental cues, we focused on cold, which is an important stress factor during early soybean growth[36–39]. We exposed soybean seedlings to control (20 °C) and cold (12 °C/5 °C day/night) conditions and harvested leaves of five-day-old seedlings every 6 min for an hour. We identified a total of 8081 phosphosites with the peptide search in MaxQuant[40], which were mapped to 3466 soybean proteins.

To identify differentially phosphorylated sites across the time course, we developed an analytical pipeline, NetPhorce, that analyzes label-free phosphoproteomics data (see "Methods"), and wrapped it into an R package (https://ksong4.github.io/NetPhorce/). The 8081 detected phosphosites were used as input for our pipeline, which first performs several quality control steps, including the removal of phosphosites with insufficient datapoints (see "Methods"). To account for technical biases and to make samples more comparable, the phosphosite intensities were normalized using variance stabilizing normalization (vsn), reducing variation between technical replicates[41]. After the quality control steps, we identified 372 high-confidence and reproducibly quantified phosphosites (localization probability ≥0.75) mapped to 320 soybean proteins. However, the phosphorylation intensities of these high-confidence phosphosites were not always detected across the entire time course. To include very lowly abundant or absent phosphosites that fall below the detection limit and thus to overcome the intrinsic detection limitation of phosphoproteomics, we included those phosphosites that were undetected in the majority or in all replicates of a sample and considered their phosphorylation absent. With this in mind, NetPhorce was set to split the collection of high-confidence phosphosites into two subsets. As a result, the first subset contained phosphosites whose intensity values were successfully quantified across the time course and were subjected to statistical analysis (see "Methods"). On the other hand, the second subset, which were here defined as absence/presence phosphosites, contained phosphosites whose intensity value was not detected at one or more time points and were not subjected to statistical analysis. Of the 372 high-confidence phosphosites, 11 were identified as absence/presence phosphosites (at least 3 out of 4 valid values in one group and absent in the other) and 310 were significantly differentially phosphorylated upon cold (at least 3 out of 4 valid values in both groups to allow statistical comparison) (Supplementary Data 7 and Supplementary Fig. 5). This prompted us to extend NetPhorce with an unbiased approach to identify from those 321 phosphosites the key regulators important for a proper cold response.

To model the regulatory interactions between kinases/phosphatases and their substrates, we developed a network inference approach that uses the predictions of PF-NET as prior knowledge. Importantly,

because this approach does not rely on known kinase motifs or protein interaction data, we could apply it to species for which such data is largely unavailable, including soybean. To detect key regulators upon cold, we mapped the causal regulatory interactions between our list of soybean kinases and phosphatases and downstream substrates. Specifically, we inferred two networks for each condition with a dynamic Bayesian network approach, available in NetPhorce. We then combined the predicted signaling pathways, visualizing condition-specific and common regulations in Cytoscape (Fig. 4 and Supplementary Data 8). Three kinases were in common between cold and controlled conditions, Glyma.19G007300 S335 (ortholog of AtTOT3, TARGET OF TEMPERATURE3), Glyma.08G037300 T220 (ortholog of AtCDKF;1), and Glyma.10G173000 S334 (ortholog of AtTOI5, TOT3-INTERACT-ING5). The membrane-associated TOT3 and TOI5 are two known regulators of moderate heat stress in *A. thaliana* and wheat, indicating

common regulatory mechanisms between cold and heat stress[42]. Interestingly, under heat stress, TOT3 was shown to play a dominant role, while our network inference predicts that upon cold stress, TOI5 is more central (Fig. 4). In addition to the highest centrality, TOI5 also showed the highest degree of rewiring upon cold stress (Fig. 4). Last, we found two regulators, Glyma.06G161200 S496 (ortholog of AtCPK4) and Glyma.07G046800 S364 (ortholog of AtSRF6) that were specific to the cold conditions. Overall, we were able to use the information extracted from PF-NET to build an analytic pipeline that allows for the annotation and exploration of signaling pathways.

**Comprehensive predictions in crops for network inference**

To enable signaling network inference in additional model and non-model species and to compare our annotations to other plant species, we used PF-NET to predict the phosphatases in four additional crops:

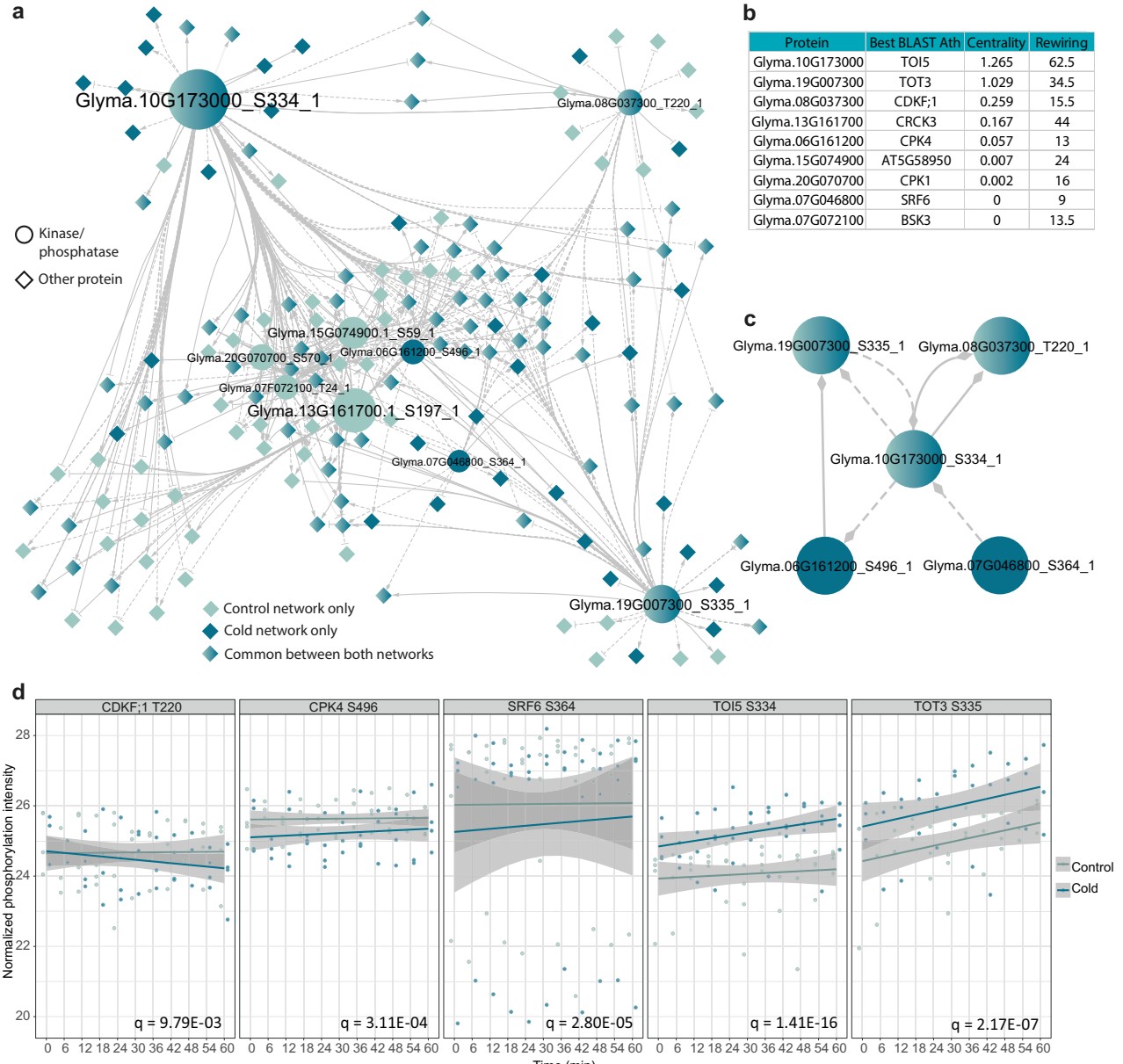

**Fig. 4 | Signaling pathways upon cold stress in soybean Altona. a** Causal relations for cold and controlled conditions were predicted with a dynamic Bayesian network approach between differentially phosphorylated kinases/phosphatases and downstream phosphosites. Gray, dark blue, and merged gray/dark blue nodes represent phosphosites present in the control, cold, or both networks, respectively. Round and triangle nodes represent kinases/phosphatases and other phosphosites, respectively. **b** The betweenness centrality and rewiring value calculated in Cytoscape for each of the upstream nodes. **c** Highlighted inferred interactions of the main nodes in the cold network. **d** Phosphorylation intensity over time of the five key nodes in the cold network. Each dot represents a replicate value.

*Triticum aestivum* (wheat), *Zea mays* (maize), *Sorghum bicolor* (sorghum), and *Oryza sativa* (rice). In total, we identified 545, 230, 170, and 180 phosphatases in wheat, maize, sorghum, and rice, respectively (Supplementary Fig. 6 and Supplementary Data 9). As for soybean, we compared our predictions with HMMER search results and the orthologs of *Arabidopsis* phosphatases. Generally, the predictions that were in common with one or two other methods had a high probability with an average of 96% (Supplementary Fig. 6). Species for which PF-NET/ HMMER would identify an increasing number of phosphatases, showed a lower/higher sequence similarity, respectively (Supplementary Fig. 7). Several proteins were functionally annotated with high probability as phosphatases solely by PF-NET, indicating that HMMER or ortholog searches by themselves are not sufficient to map an entire protein family, as we showed for the soybean kinases.

To gain insights into phosphatase functions and evolution, we combined our predictions for each species into one comprehensive list and performed a proteome-wide phylogenetic analysis (see "Methods"). The phylogenetic analysis of all six species, including *A. thaliana* and soybean, assigned our phosphatases to 229 orthogroups, i.e., the

generalization of orthology to multiple species (Fig. 5a, b and Supplementary Data 10). In total, 61 of those orthogroups contained 10 or more phosphatases, in total accounting for ~60% of the predicted phosphatases (Fig. 5a and Supplementary Data 10). 73 orthogroups containing 1011 (63.5%) phosphatases were shared among all six species, which suggests these phosphatases might play roles essential to the plant's fitness (Fig. 5b). As expected, the first bipartition in the phylogenetic species tree separates dicots and monocots (Fig. 5c). Another 30 and 8 orthogroups are shared only among the monocots and dicots, respectively, suggesting that phosphatases in these groups have functions specific to this evolutionary divergence (Fig. 5b)[43]. Only 80 phosphatases were not assigned to an orthogroup, indicating high representation of orthology relationships within the entire proteomes. Another 55 phosphatases did not have orthology relationships with other predicted phosphatases, which suggest these could be false positives[44]. Overall, this phylogenetic analysis showed that most phosphatases are shared among our six species and confirmed the functional annotation of 91.5% of the phosphatases. Similar as for soybean, a comprehensive list of regulatory proteins in wheat, rice,

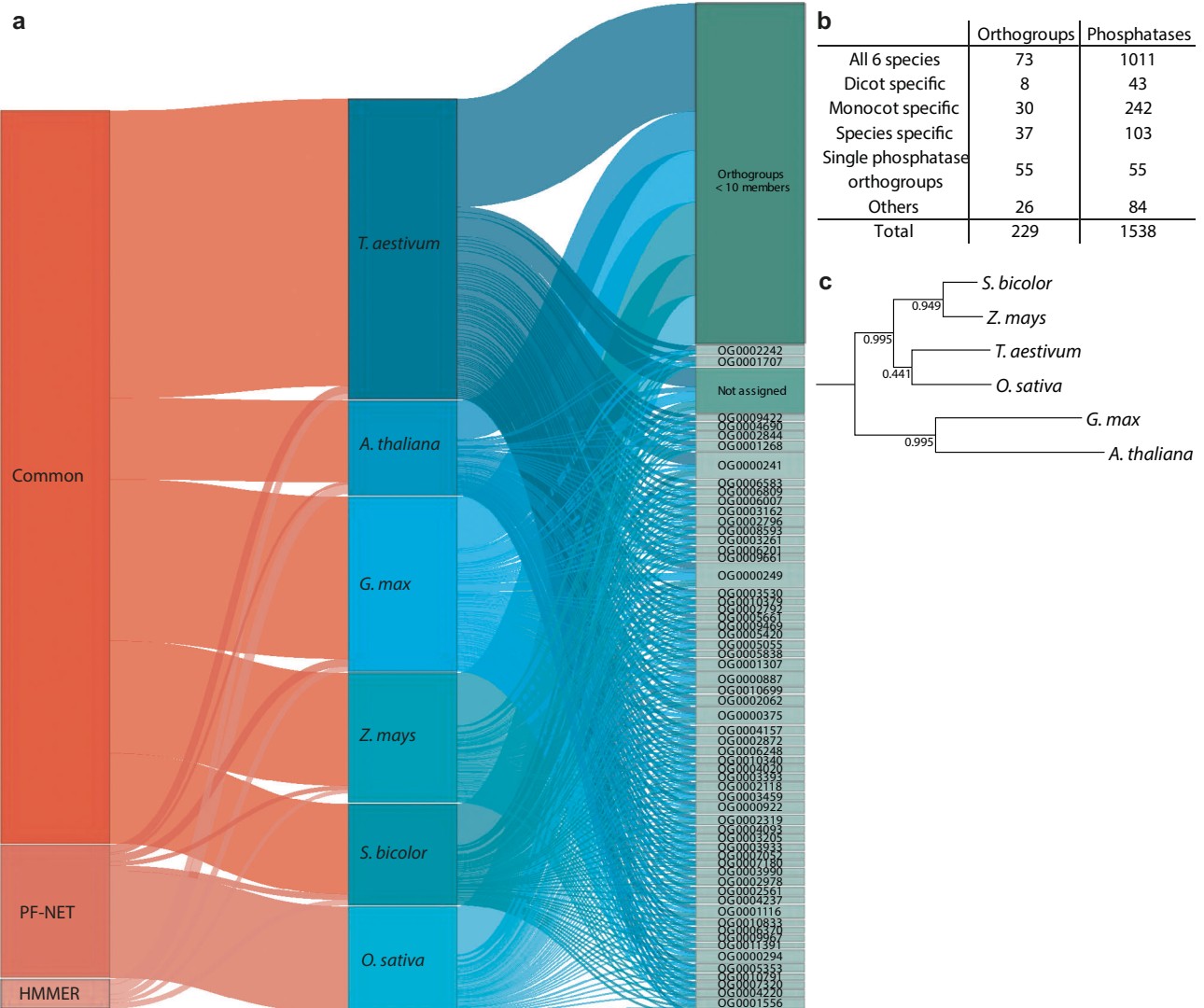

**Fig. 5 | Evolutionary conservedness of predicted phosphatases in *A. thaliana*, *Glycine max* (soybean), *Triticum aestivum* (wheat), *Zea mays* (maize), *Sorghum bicolor* (sorghum), and *Oryza sativa* (rice). a** Sankey diagram visualizing the predicted phosphatases. The width of the connections between each vertical block represents the number of phosphatases (first block) predicted with PF-NET, HMMER, or both ("Common"), (second block) within each species, and (third block) within each orthogroup. **b** Number of inferred orthogroups and phosphatases of the six species. **c** Inferred phylogenetic species tree. The support values for each bipartition indicate the proportion of times that the bipartition is observed in the individual species tree estimates.

sorghum, and maize, would ensure the possibility for signaling networks inference. To this end, we also predicted the kinases in those species using PF-NET and HMMs. We included our phosphatase and kinase predictions, as well as published predictions[45] of another 20 species as a default table in the NetPhorce R package, enabling signaling network inference in non-model species. Overall, the combined use of computational tools leads to highly confident predictions, and PF-NET can identify functional proteins that are otherwise missed by existing methods.

## Discussion

Applying neural networks to classification problems is highly promising in molecular biology[5, 7,18,46]. Here, we applied a neural network to classify proteins into one of almost 1000 protein families based on their amino acid sequence and predicted the kinases and phosphatases for six species, including soybean, wheat, rice, sorghum, maize, and *A. thaliana*. By identifying these regulatory proteins, we were able to perform a network biology study in soybean and shed light on the signaling cascades underlying cold stress. Using our approach, we identified a potential common regulatory mechanism between cold and heat stress, in which TOT3 and TOI5 are the key nodes. These results suggest that TOT3 and TOI5 could function as thermostats, phosphorylating a different set of substrates depending on the temperature. Thus, next studies will focus on validating predicted targets under temperature gradients to elucidate the precise molecular mechanisms of TOT3 and TOI5 in regulating temperature responses. Additionally, we identified two cold-specific regulators, Glyma.06G161200 and Glyma.07G046800. In *A. thaliana*, the ortholog of Glyma.06G161200, AtCPK4, is a positive regulator of ABA signaling, a key hormone mediating plant responses to various abiotic stresses, including cold[47]. Moreover, the rice ortholog of Glyma.06G161200, OsCPK24 (Os11g0171500), was shown to play a role in cold stress responses[48]. Thus, our approach offers the possibility to discover central kinases that could be candidates for genome editing strategies.

Mapping the function of proteins by classifying them into their respective family provides clues to their structure, localization, and activity, and is thus essential to understand molecular responses and phenotypes. Deep learning provides a framework to achieve this task and classify proteins into their families without prior knowledge of sequence features. Our neural network approach can identify distant dependencies and higher-order correlations within the full sequence. PF-NET was able to classify unknown and uncharacterized proteins to the kinase and phosphatase families. For example, CEX1 was predicted to function as a kinase solely by PF-NET. Identifying kinases/phosphatases impacts downstream analysis as the inclusivity of regulators reduces uncertainties and increases overall accuracy for network inference. Omitting or not considering predicted regulators would lead to incomplete predictions and systemic errors. In addition, many target proteins can be (de)phosphorylated by multiple upstream kinases/phosphatases, resulting in a complex regulatory module for every target protein. Identifying the full extent of this plethora of upstream regulators is thus critical to fully map the regulatory modules of a protein's phosphorylation events and, concordantly, the complexity of signaling cascades.

The newly annotated proteins were missed by previous computational methods, potentially because PF-NET is capable of identifying nonlinearity and amino acid dependencies across the full sequence. Thus, a combination of annotation approaches greatly increases coverage and reduces errors. Recently, deep learning has been implemented to expand the coverage of the Pfam database and further annotate unknown proteins[16]. In addition, intuitive and conditional thresholding can be applied depending on the desired stringency. Thresholding is necessary because PF-NET is configured to have 996 output classes and categorizes every sequence even if the sequence does not belong to one of these 996 classes. Adding a threshold ensures highly accurate predictions. To enable easy thresholding and fast in bulk predictions of a collected set of sequences or a reference proteome of interest, we made our neural network available through our easy-to-use webtool (https://sozzanilab.shinyapps.io/PF-NET_Shiny/). This webtool greatly promotes accessibility and interpretability. Last, with some bioinformatics knowledge, PF-NET is retrainable to add protein families or update current families, which greatly contributes to its applicability beyond the identification of kinases and phosphatase. For example, PF-NET has a high predictive power (f1 score >90%) for several transcription factor families, including bHLH, MYB, and zinc-finger transcription factors.

To explore signaling cascades, we developed a data-driven network inference approach. To this end, our approach does not rely on known substrates of kinases, protein-protein interaction data, or consensus sequence motifs, which leads to unbiased identification of phosphorylation dynamics. This approach uses dynamic Bayesian principles[21] and requires only two sources of information: (i) a time-course phosphoproteomics dataset, and (ii) a list of kinase/phosphatase proteins for that species. Combined with PF-NET, our approach can be used to study signaling mechanisms in any species. To facilitate the dissemination within the scientific community, we wrapped our phosphoproteome analysis and network inference pipeline in an R package, which can be used by biologists with minimal programming and bioinformatics knowledge.

## Methods
### Training dataset and encoding
The Pfam database contains >19,000 protein families. To obtain a number of protein families adequate to handle and train the neural network, we selected a total of 996 protein families (Supplementary Data 2) by excluding protein families with fewer than 100 annotated sequences within Pfam's underlying sequence database and focusing on protein families within the plant and animal kingdoms. All sequences found in Pfam's underlying sequence database of each Pfam family were extracted using MySQL workbench. The script used is available at https://github.com/LisaVdB/PF-NET. We then excluded long sequences at a cutoff of 1234 amino acids based on the sequence length distribution. As such, the total training dataset consisted of 7,385,028 sequences, covering the entire tree of life. This large database of sequences was used as input for the neural network without any feature extraction.

Protein sequences are composed from 20 amino acids, each represented by a letter, according to the IUPAC guidelines. An additional four letters are used to represent a group of amino acids: Z represents a glutamic acid or glutamine; letters O and U represent pyrrolysine and selenocysteine, respectively; and letter X represents any amino acid. Protein sequences were encoded using a binary encoding scheme, in which each amino acid is represented as a string of zeros and ones with length 5 (Fig. 1). Binary encoding has the advantage that the number of input columns (5 columns for binary encoding versus 21 columns for one-hot encoding) and thus dimensionality of the dataset requires less memory. The binary encoding scheme requires 128GB RAM to load. The sequences were zero-padded until length 1234 to preserve the original input size.

After binary encoding of the sequences and to facilitate training, the dataset was split into 6 batches of around 1.2 million sequences in a stratified manner, i.e., each batch contained a similar number of sequences for each of the 996 families. Three batches were combined to make 5 different batch combinations, each of around 3.6 million sequences. Subsequently, each batch was split in a stratified manner into three proportions: 60% of sequences are used for training, 20% for model validation, and 20% are held out as test sequences. Stratified train-test-split is performed to ensure all classes are divided proportionally into the training, validation, and testing sets[49].

## Neural network architecture

The neural network architecture consists of four different layers: (1) a convolutional neural network layer (CNN), (2) an attention layer, (3) a biLSTM layer, and (4) two dense layers connected to the output vector. To extract putative protein domains within the sequence, we added a 1D CNN that performs a convolution across the encoded sequences with a kernel size of 7 (Fig. 1a). Smaller sequence domains conserved within a protein family can thus be identified with the CNN. The kernel moves along the sequence with a stride of 1 to compute features. A total of 320 filters are applied, leading to the computation of 320 different features (Fig. 1a). To emphasize the learned patterns from the CNN layer, we included an attention layer that integrates the entire sequence generated from the CNN layer, and generates an output sequence that is a function of the entire input sequence and all its hidden states. The function is an "attention" operation, giving different weights to each hidden state (Fig. 1a). As such, the attention layer will assign increased importance to key domains within the sequence. To capture long-distance dependencies within the sequences and between detected domains, we added a biLSTM layer. The biLSTM layer processes the output of the attention layer to identify any distant dependencies within the entire sequence (Fig. 1a). As such, the order of distant conserved sequence domains is taken into account. Lastly, two dense layers are connected to the output vector (Fig. 1a).

## Experimental settings and evaluation

The hyperparameters in Supplementary Table 1 were found to give the best performance. Several learning rates for the Nadam (Nesterov-accelerated Adaptive Moment Estimation) were tested ($1E^{-3}$, $1E^{-5}$, $1E^{-7}$)[50]. A learning rate of $1E^{-7}$ in combination with a learning rate decay with every epoch gave the best performance with respect to the loss. Additionally, three different values of the number of filters (128, 320, and 480) for the CNN were tested, of which 320 filters gave the best performance with respect to the loss. Two activation functions (softmax, sigmoid) at the final layer were tested of which softmax led to the best performance with respect to the loss[51,52].

The input dataset from the Pfam database is unbalanced, meaning that the number of sequences for each protein family (i.e., the class size) is unequal. Thus, class sizes are skewed: 21.5% protein families are represented by <1000 sequences, 61.8% have >1000 and <10,000 sequences, and 16.7% have >10,000 sequences. To accurately predict such an unbalanced dataset and correct for skewed class sizes, we applied a focal loss function to evaluate the prediction error of PF-NET[53]. Lastly, to take into account the skewed distribution of the classes, different class weights were assigned to the classes when fitting the model that assign different costs to misclassification according to their sample distribution (i.e., misclassification of minority classes with lesser samples are penalized higher than majority classes).

To evaluate the performance of the neural network, the precision, recall, and f1 score of each class individually was calculated as follows:

$$Precision = \frac{True\ positives}{True\ positives + False\ positives} \qquad (1)$$

$$Recall = \frac{True\ positives}{True\ positives + False\ Negatives} \qquad (2)$$

$$f1\ score = \frac{2 \times Precision \times Recall}{Precision + Recall} \qquad (3)$$

To evaluate the overall performance of the neural network, the accuracy as well as the macro average and weighted average of precision, recall, and f1 score were calculated. For the macro average, the scoring metrics of each class were calculated and averaged. For the weighted average, the scoring metrics for each class were calculated, weighted by the number of true instances for each label, and averaged to account for class imbalance.

All scripts for the neural network training, validation, and for making predictions are available on GitHub (https://github.com/LisaVdB/PF-NET). A webtool was also built to easily make predictions with PF-NET (https://sozzanilab.shinyapps.io/PF-NET_Shiny/).

## Neural network evaluation

To set up and benchmark our interpretable and robust pipeline, we used the yeast proteome as a benchmark independent dataset. The yeast proteome was downloaded from the yeast genome database (*Saccharomyces cerevisiae* reference strain S288C release R64-3-1 2021-04-21, Taxon identifier 559292). Sequences larger than 1234 amino acids were split into sequences of 1234 amino acids or smaller. Sequences were encoded and predictions were made with PF-NET. The predictions were compared against a carefully curated list of experimentally validated kinases and phosphatases from published literature, which we refer to as the ground truth (Supplementary Data 3). For comparison with PF-NET, hidden Markov models of the Pfam families Metallophos (PF00149), DSP (PF00782), PP2C (PF00481), Phosphatase_Tyr (PF00102), and Pkinase (PF00069) were used to perform a HMMER search (https://www.ebi.ac.uk/Tools/hmmer/search/hmmsearch). We used the default values (0.01 and 0.03 E value cutoffs) and restricted the HMMER search by taxonomy (organism: *S. cerevisiae*).

To identify a threshold that minimizes false positives and false negatives, 100 thresholds between 0 and 1 were tested. The cost associated with each threshold was calculated as follows.

$$Cost = FDR - (3 * TPR) \qquad (4)$$

$$FDR = False\ positives/(False\ positives + True\ positives) \qquad (5)$$

$$TPR = True\ positives/(True\ positives + False\ Negatives) \qquad (6)$$

The threshold with the lowest cost was selected. The R script for threshold identification is available at GitHub (https://github.com/LisaVdB/PF-NET). All plots are generated in Microsoft Excel or R (version 4.2.2)[54] using ggplot2 (version 3.4.1)[55], or networkD3 (version 0.4) and htmlwidgets (version 1.6.1) for the Sankey diagram.

## Neural network predictions in plants

To test whether our neural network will generalize well toward plant proteins, we made functional predictions in several model and non-model plant species: *A. thaliana*, soybean, wheat, maize, sorghum, and rice. After downloading the proteomes of these species in their respective databases, sequences larger than 1234 amino acids were split into sequences of 1234 amino acids or smaller. Then, PF-NET was used to make functional predictions. The *A. thaliana* proteome was downloaded from TAIR (Arabidopsis thaliana Genome Annotation Official Release version Araport11, release date June 2016). The soybean proteome was retrieved from Soybase (Williams 82 Genome Sequencing Project, Assembly 2 Annotation 1). The entire proteome of *Zea mays* (maize) (B73 reference NAM v5.0), *Sorghum bicolor* (sorghum) (Sbi1.4/SbGDB181), *Triticum aestivum* (wheat) (IWGSC RefSeq v2.1 High Confidence peptides), and *Oryza sativa* (rice) (v7.0) were downloaded and tested.

We selected the kinases from *A. thaliana* and soybean by identifying all protein sequences that were predicted to be part of the Pfam families Pkinase (PF00069) and Pkinase_Tyr (PF07714). To identify the phosphatase of *A. thaliana*, soybean, rice, wheat, maize, and sorghum at a proteome level, the following four Pfam families were selected: Metallophos (PF00149), DSP (PF00782), PP2C (PF00481), and Phosphatase_Tyr (PF00102).

To test the generalization of our neural network, we compared PF-NET's results to published computational protein classifications studies and HMMER runs. We did this specifically for the *Arabidopsis thaliana* kinases and phosphatases, and the *Glycine max* kinases. Similarly, as for *Saccharomyces cerevisiae*, the cost was calculated for 100 probability thresholds for the predicted *Arabidopsis thaliana* kinases and phosphatases, and the *Glycine max* kinases, using the literature-obtained kinases and phosphatases as ground truth. This led to the identification of an optimal threshold of 0.616 for *A. thaliana* kinases, 0.565 for the *A. thaliana* phosphatases, and 0.646 for the soybean kinases. *A. thaliana* and soybean kinases and *A. thaliana* phosphatases were retrieved from literature studies[29,30,56]. Hidden Markov models of the Pfam families Pkinase (PF00069) and Pkinase_Tyr (PF07714) were used to search for kinases with HMMER (https://www.ebi.ac.uk/Tools/hmmer/search/hmmsearch) (HmmerWeb version 2.41.2). We used the default values (0.01 and 0.03 *E* value cutoffs) and restricted the HMMER search by taxonomy (species-level). Gene descriptions for *A. thaliana* were downloaded from TAIR. Gene descriptions, GO-terms, and Panther descriptions for soybean were downloaded from SoyBase[57].

To annotate the undescribed phosphatases in other non-model species, we made predictions as described above. To identify the phosphatases of the other crops, we used a mild threshold of 0.5. Neural network predictions were compared with a HMMER search and soybean, maize, sorghum, wheat, and rice orthologs of known *A. thaliana* phosphatases. Hidden Markov models of the Pfam families Metallophos (PF00149), DSP (PF00782), PP2C (PF00481), and Phosphatase_Tyr (PF00102) were used to search for phosphatases with HMMER (https://www.ebi.ac.uk/Tools/hmmer/search/hmmsearch). Orthologs were identified with PLAZA 4.5 through the Plaza Integrative Method (https://bioinformatics.psb.ugent.be/plaza/)[58]. To infer the orthogroups for our six species and a rooted species tree, we performed a comprehensive phylogenetic analysis with the six species using Orthofinder (2.5.4), a software program for phylogenetic orthology inference[59,60]. To quantify the sequence percent identity among the phosphatases of each plant species, we extracted the amino acid sequences and used those as input for Clustal Omega, a multiple sequence alignment program[61]. Clustal Omega results provide the sequence percent identity matrix.

## Label-free phosphoproteomics

Four Altona[62] soybean seeds (USDA-ARS Germplasm Resource Information Network nr: PI 548504) were sown at 2 cm depth in one square plastic pot (7 × 7 × 8 cm) filled with regular potting soil (Beroepspotgrond, Saniflor). After 10 days, seedlings were thinned to two per pot. The pots were placed in a controlled growth room with 400 μmol m$^{-2}$ s$^{-1}$ light intensity, continuous 20 °C, and 15 h/9 h dark day/night rhythm. Regular watering was performed to keep the plants under optimal growing conditions, no extra fertilizer was provided as a base fertilizer (NPK 12-14-24 with micronutrients at 1.2 kg m$^{-3}$) was present in the potting soil. A total of 60 pots per treatment were sown. For the cold treatment, the seedlings were transferred five days after germination to 12 °C/5 °C. The plants were kept in another growth chamber at 20 °C during the night before the start of the cold treatment. The original growth chamber was set at 12 °C/5 °C during that night. The cold treatment started by transferring the plants into the original growth chamber at the moment that the lights were switched on in the morning time (at a temperature of 10 °C). Every 6 min up to one hour after treatment, the tip half of unifoliate leaves from three distinct plants were pooled for phosphoproteomics in 5-mL tubes containing 4× metal beads of 4 mm, followed by snap-freeze in liquid nitrogen. The process was repeated four times for four biological replicates.

To extract proteins, the frozen material of each pool was grounded to fine powder. After, the material was resuspended in

homogenization buffer containing 30% sucrose, 50 mM Tris-HCl buffer (pH 8), 0.1 M KCl, 5 mM EDTA, and 1 mM DTT in Milli-Q water, to which one tablet of both cOmplete™ Ultra protease inhibitor mixture (Roche) and PhosSTOP phosphatase inhibitor mixture (Roche) were added. The samples were sonicated on ice to further break cells and subcellular organelles and centrifuged at 4 °C for 15 min at 3200 g to remove debris. Supernatants were collected and a methanol/chloroform precipitation was carried out by adding 3, 1, and 4 volumes of methanol, chloroform, and water, respectively. Samples were centrifuged for 10 min at 3200 × *g* and the aqueous phase was removed. After the addition of four volumes of methanol, the proteins were pelleted by centrifugation for 10 min at 3200× *g*. Pellets were washed with 80% acetone, and centrifuged for 10 min at room temperature at 3200 × *g*. The supernatants were discarded, and the pellets were left to dry on air. Protein pellets were then resuspended in 8 M ureum in 50 mM triethylammonium bicarbonate (TEAB, Sigma-Aldrich) buffer (pH 8). Alkylation of cysteines was carried out by adding tris(carboxyethyl)phosphine (TCEP, Pierce) and iodoacetamide (Sigma-Aldrich) to final concentrations of 15 mM and 30 mM, respectively, and the samples were incubated for 15 min at 30 °C in the dark. Three mg of protein material was pre-digested with 10 μg of MS-grade lysyl endopeptidase (Wako Chemicals) for 2:30 min at 37 °C. The mixtures were diluted eightfold with 50 mM TEAB, followed by an overnight digestion with trypsin (Promega) with an enzyme-to-substrate ratio of 1:100. The digest was acidified to pH 3 with trifluoroacetic acid (TFA, Biosolve) and desalted using SampliQ C18 SPE cartridges (Agilent) according to the manufacturer's guidelines. For phosphopeptide enrichment, the desalted peptides were fully dried in a vacuum centrifuge and then washed in 500 μl of loading buffer [80% (v/v) acetonitrile (BioSolve), 6% (v/v) TFA] with gentle agitation (800 rpm). Briefly, the resuspended peptides were incubated with 1 mg MagReSyn® Ti-IMAC (ReSyn Biosciences) microspheres for 20 min at room temperature with continuous mixing. After, the tubes were placed to a magnetic separator for 10 s having their supernatant removed and discarded after. Following, the microspheres were washed once more with loading buffer, followed by wash solvent 1 (60% acetonitrile, 1% TFA, 200 mM NaCl (Sigma-Aldrich)) and twice with wash buffer 2 (60% acetonitrile, 1% TFA). The bound phosphopeptides were eluted with three volumes (80 μl) of elution buffer (40% acetonitrile, 1% NH4OH (S)), immediately followed by acidification to pH 3 using 6 μl 100% formic acid (Roche). Prior to MS analysis, the samples were vacuum dried and re-dissolved in 50 μl of 2% (v/v) acetonitrile and 0.1% (v/v) TFA.

Peptides were re-dissolved in 50 μl loading solvent A (0.1% TFA in water/ACN (98:2, v/v)) of which 3 μl was injected for LC-MS/MS analysis on an an Ultimate 3000 RSLC nano LC (Thermo Fisher Scientific) in-line connected to a Q Exactive mass spectrometer (Thermo Fisher Scientific). The peptides were first loaded on a μPAC™ Trapping column with C18-endcapped functionality (Pharmafluidics) and after flushing from the trapping column the peptides were separated on a 50 cm μPAC™ column with C18-endcapped functionality (Pharmafluidics) kept at a constant temperature of 35 °C. Peptides were eluted by a linear gradient from 98% solvent A' (0.1% formic acid in water) to 55% solvent B' (0.1% formic acid in water/acetonitrile, 20/80 (v/v)) in 120 min at a flow rate of 300 nL/min, followed by a 5 min wash reaching 99% solvent B'.

The mass spectrometer was operated in data-dependent, positive ionization mode, automatically switching between MS and MS/MS acquisition for the 5 most abundant peaks in a given MS spectrum. The source voltage was set at 3 kV and the capillary temperature at 275 °C. One MS1 scan (*m/z* 400–2000, AGC target 3 × 10$^6$ ions, maximum ion injection time 80 ms), acquired at a resolution of 70,000 (at 200 *m/z*), was followed by up to 5 tandem MS scans (resolution 17,500 at 200 *m/z*) of the most intense ions fulfilling predefined selection criteria (AGC target 5 × 10$^4$ ions, maximum ion injection time 80 ms,

isolation window 2 Da, fixed first mass 140 $m/z$, spectrum data type: centroid, under-fill ratio 2%, intensity threshold 1.3xE4, exclusion of unassigned, 1, 5–8, >8 positively charged precursors, peptide match preferred, exclude isotopes on, dynamic exclusion time 12 s). The HCD collision energy was set to 25% Normalized Collision Energy, and the polydimethylcyclosiloxane background ion at 445.120025 Da was used for internal calibration (lock mass).

MS/MS spectra files were searched against the Soybean database (Williams 82 Genome Sequencing Project, Assembly 4 Annotation 1) with Maxquant software version 1.6.10.43, a program package that allows MS1-based label-free quantification[40,63]. Searches were performed within replicates (0, 6, 12, 16, 24, 30, 36, 42, 48, 54, and 60 min samples) with both control and cold treatment groups, with "match between runs" feature enabled in order to maximize peptide identification. Next, the four "Phospho(STY).txt" output files from the four replicates were merged into a single file. For that, if two or more replicates shared same values for all columns: "protein id", "position", "aminoacid" and "multiplicity", the quantification values were merged to a pre-existing row; if not, it was appended to the dataframe as a new row. For downstream analysis only the merged file was used. The mass spectrometry proteomics data, the MaxQuant settings, MaxQuant outputs and resulting merged file have been deposited to the ProteomeXchange Consortium via the PRIDE[64] partner repository with the dataset identifier PXD037601.

## Analysis and network inference with NetPhorce

Prior to statistical analysis, data cleaning and quality controls were performed, which includes several filtering steps, missing value handling, and normalization. Specifically, contaminates and reverse peptides identified by MaxQuant were removed, and phosphosites with a localization probability below 0.75 were removed. Generally, two types of missingness can be found within (phospho)proteomics: (1) missingness completely at random (MCAR), and (2) missingness not at random (MNAR)[65]. MCAR occurs due to technical or instrumentation defects, such as poor ionization, other peptides competing for charge, and enzymatic modifications, and is unrelated to its abundance. On the other hand, MNAR is related to the peptide's abundance and occurs when a peptide abundance falls below the instrument detection limit or a peptide is simply not present. Phosphopeptides that are very lowly abundant or not present in one condition versus abundant in another condition (e.g., cold versus control conditions) are biologically relevant as they can, for example, be part of a pathway that is only activated upon cold. In our approach, we included MNAR and discarded MCAR, hence including the biologically relevant peptides but excluding the peptides with too few datapoints. To ensure a balanced dataset for statistical analysis, a threshold of 3 or more valid values or missing values (zeros) per replicate for each phosphosite across the time course and experiment was chosen. Phosphosites that do not meet these criteria were filtered out. For example, a phosphosite with 2 valid values and 2 zeros at one of the time points was removed. A phosphosite with 1 valid value and 3 zeros or with 4 zeros at one or more time points and 3 or more valid values for the other time points was retained and classified as an absent/present phosphosites, which was not subjected to statistical analysis. A phosphosite with 3 or more valid values at all time points was retained and subjected to statistical analysis.

Next, to normally distribute the intensity values, they were $\log_2$ transformed. Variance stabilizing normalization (vsn) was performed to reduce the variation between replicates[41,66]. To identify phosphosites that were differentially phosphorylated between conditions or time points, a linear mixed model was fitted to the phosphoproteomics data depending on the experimental design as follows:

$$Y = \begin{cases} \mu + \alpha_i + \gamma_k + \varepsilon & \text{if } n=1 \ \& \ t>1 \\ \mu + \beta_j + \gamma_k + \varepsilon & \text{if } n>1 \ \& \ t=1 \\ \mu + \alpha_i + \beta_j + \alpha\beta_{ij} + \gamma_k + \varepsilon & \text{if } n>1 \ \& \ t>1 \end{cases} \tag{7}$$

Where $Y$ is the phosphorylation intensity, $\alpha_i$ is a fixed effect for the condition variable, $\beta_j$ is a fixed effect for the time variable, $\gamma_k$ is a random effect for the replicate variable, and $\varepsilon$ is the within-replicate error. A random effect for replicate was included in the model to account for correlation between plants grown at the same time. As our objective was to identify factors key for the cold response in soybean seedlings, we selected the significantly differentially phosphorylated peptides upon cold (i.e., a $P$ value of the condition variable <0.05). Generally, NetPhorce will select the $p$ value of the condition variable unless the input data does not contain a condition variable, in that case, NetPhorce will consider the $P$ value of the time variable. To handle the multiple hypothesis-testing problem, which leads to an increased probability of identifying significant hits, the q-values were estimated from the calculated $P$ values[67]. A $q$-value cutoff of 0.05 was chosen. All intermediate steps of our pipeline were benchmarked against the Perseus software platform[68] and showed the same results (Supplementary Table 2). Phosphosites that were significantly differentially phosphorylated between the control and cold conditions, as well as the absent/present phosphosites were selected for further network inference.

The network inference was based on dynamic Bayesian network principles and consisted of three major steps: (1) identifying potential regulator-target combinations, (2) scoring these regulations according to Bayesian principles, (3) selecting high-scoring regulations and determining their sign. To infer the regulatory interactions, we leveraged the changes over time for each treatment, allowing us to probabilistically calculate the Bayesian score, as follows.

The median was calculated for each condition and subtracted from intensity values to retrieve median-centered $\log^2$ transformed data. Based on the time series phosphoproteome data, changes in phosphorylation intensities were identified. The delta phosphorylation events of a protein ($p$) are calculated as follows:

$$p_\Delta(t) = \begin{cases} 0, & |p(t) - p(t-1)| < \text{quantile}(\forall \, |p(t) - p(t-1)|, c) \\ 1, & p(t) - p(t-1) > a \cdot p(t-1) \\ -1, & p(t) - p(t-1) < -b \cdot p(t-1) \end{cases} \tag{8}$$

Specifically, an increase or decrease in phosphorylation intensity between two consecutive time points was identified when the intensity increased or decreased with at least 25%, respectively. A bottom percentage of all fold changes equal to 10% was considered as unchanged. Absent phosphorylation intensities are set to 0. Next, potential regulator ($r$) – target ($p$) combinations were identified as follows:

$$\left( \frac{\sum_{t=1}^{n} r_\Delta(t) = p_\Delta(t)}{n} \right) \bigg| \left( \frac{\sum_{t=1}^{n} r_\Delta(t-1) = p_\Delta(t)}{n-1} \right) > 0.5 \tag{9}$$

$$r_\Delta(t) = r(t) - r(t-1) \tag{10}$$

$$p_\Delta(t) = p(t) - p(t-1) \tag{11}$$

Where $n$ is the number of time points. Only identified kinases and phosphatases from HMMER and PF-NET predictions (analyses were redone using the latest soybean genome annotation Williams 82 Genome Sequencing Project, Assembly 4 Annotation 1) were considered as potential regulators. A protein is a potential regulator of a target protein if and only if it exhibits a change in phosphorylation

intensity at the same time (time-lapse 0) or immediately prior (time-lapse 1) to a change in phosphorylation intensity of the target for at least 50% of the time points[21]. Given our time course (every 6 min for one hour) and the fast nature of phosphorylation cascades, we selected time-lapse 0 and 1 as we assumed that the regulation may happen within and between time points. To substantiate the choice of including both time-lapse 0 and 1, we also proceeded with only time-lapse 1, compared network topology, and were able to draw the same conclusions (Supplemental Fig. S8). For time-course datasets with smaller time steps, only the inclusion of time-lapse 1 is advised[22,23]. NetPhorce R package contains a default list of kinases and phosphatases for 26 species. For rice, soybean, wheat, sorghum, arabidopsis, and maize the kinases and phosphatases from the union of PF-NET's, HMMERs, and/or orthologs results were included.

Next, the data were discretized into three levels: (1) a discretization level below the experiment-determined median, (2) a discretization level above the experiment-determined median, and (3) a discretization level for absent phosphorylation intensities. Inferring a DBN consists of finding the network topology that maximizes a score, i.e., finding the most likely parents of each node. Each potential regulator-target or regulator 1 & 2-target combination were scored with the Bayesian Dirichlet equivalent uniform (BDeu)[21]. The BDeu score of a DBN can be decomposed as the sum of the scores of the log conditional probabilities of each node[21]:

$$BDeu(D, G) = \sum_{i=1}^{n} \sum_{j=1}^{q_i} \left( \log \left( \frac{\Gamma(\frac{\alpha}{q_i})}{\Gamma(\sum_{k=1}^{r_i} N_{ijk} + \frac{\alpha}{q_i})} \right) + \sum_{k=1}^{r_i} \log \left( \frac{\Gamma(N_{ijk} + \frac{\alpha}{q_i})}{\Gamma(\frac{\alpha}{r_i q_i})} \right) \right)$$

(12)

Where $G$ refers to the Bayesian graph, $D$ refers to the dataset containing the time point observations, $N_{ijk}$ indicates the number of data vectors in which target $i$, has the value $k$ while its parents are in configuration $j$. $\alpha$ equals $1E^{-15}$, a hyperparameter of the Dirichlet distribution. The regulators of target $i$ are the ones that led to the highest value of the BDeu.

For each inferred edge, a score is calculated to determine whether the inferred interaction is a phosphorylation or dephosphorylation. For time-lapse 0, the change over time of the target and regulator was compared at the same time points, while for time-lapse 1, the change over time of the target was compared with that of the regulator at the immediate prior time point. If the change over time is in the same direction (for example, an increase in phosphorylation) for most of the time points, the direction is considered phosphorylation. While if intensities of the regulator and target are changing in opposite directions (for example, one increases in phosphorylation while the other decreases in phosphorylation), the direction is considered dephosphorylation. If the change over time is equal, then the sign of the interaction is denoted as undetermined.

All networks were visualized in Cytoscape® 3.8.0[69]. To evaluate how the control and cold signaling networks were rewired, we used DyNet, a Cytoscape application that, among others, allows for the analysis of the most "rewired" nodes across networks with the node attribute "DyNet REWIRING"[70].

### Reporting summary
Further information on research design is available in the Nature Portfolio Reporting Summary linked to this article.

## Data availability
The mass spectrometry proteomics data, the MaxQuant settings, MaxQuant outputs, and the resulting merged file generated in this study have been deposited in the ProteomeXchange Consortium via the PRIDE[64] database under accession code PXD037601. The prediction data, PF-NET's performance, network inference output, and orthogroup data generated in this study are provided in the Supplementary Information. Sequences from Pfam's underlying sequence database of each Pfam family used in this study were extracted from http://ftp.ebi.ac.uk/pub/databases/Pfam/. The yeast, *A. thaliana*, soybean, wheat, sorghum, rice, and maize proteome used in this study were downloaded at the Saccharomyces Genome Database (https://www.yeastgenome.org/), TAIR (https://arabidopsis.org/), Soybase (https://www.soybase.org/), EnsemblPlants (https://plants.ensembl.org/Triticum_aestivum/Info/Index), EnsemblPlants (https://plants.ensembl.org/Sorghum_bicolor/Info/Index), rice genome annotation project (http://rice.uga.edu/), and Maize Genome Database (https://www.maizegdb.org/).

## Code availability
All scripts for the neural network training, validation, and for making predictions are available on GitHub (https://github.com/LisaVdB/PF-NET, https://doi.org/10.5281/zenodo.8047493). A webtool was also built to easily make predictions with PF-NET (https://sozzanilab.shinyapps.io/PF-NET_Shiny/). All scripts to analyze label-free phosphoproteomics data with NetPhorce were wrapped into an R package (https://ksong4.github.io/NetPhorce/).

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

## Acknowledgements

This work was supported by the Foundation for Food and Agriculture Research (FFAR CA18-SS-0000000026), Benson Hill, VIB, BASF, the United Soybean Board (2020-152-0134), and the North Carolina Soybean Producers Association (20-122) to R.S., A.M.L., and I.D.S., the National Science Foundation (NSF) (PGRP BIO-2112058) to R.S., and the Research Foundation—Flanders (FWO.OPR.2019.0009.01) to I.D.S.

## Author contributions

L.V.d.B.: conceptualization, data curation, formal analysis, investigation, methodology, software, validation, visualization, and roles/writing—original draft. D.K.B.: formal analysis, investigation, methodology, validation, and writing—review and editing. K.S.: software. C.F.F.d.L.: formal analysis, writing—review and editing. M.A.: formal analysis, and writing—review and editing. P.N.: investigation. T.S.: investigation. T.Z.: investigation. S.Z.: investigation. A.C.O.: investigation. J.A.: investigation. P.L.: investigation. B.V.D.C.: investigation. I.D.S.: conceptualization, funding acquisition, writing—review and editing. A.M.L.: conceptualization, funding acquisition, and writing—review and editing. R.S.: conceptualization, funding acquisition, supervision, and writing—review and editing.

## Competing interests

The authors declare no competing interests.
