## [Peer Review File · Nature Communications]

Functional annotation of proteins for signaling network inference in non-model speciesREVIEWER COMMENTS

Reviewer #1 (Remarks to the Author):

The authors presented a multi-layer neural network consisting of convolutional neural network (CNN) and bidirectional long short term memory (biLSTM) to construct feature representations and classify proteins into 996 protein families for protein function annotation. Moreover, a network inference method with a dynamic Bayesian network was developed to predict signaling cascades. The authors conducted adequate experiments on six species including *Arabidopsis thaliana*, soybean, wheat, maize, rice, and sorghum, and the results demonstrated that the proposed method is highly generalizable and applicable in protein function annotation. However, taking care of below points, will make the paper more informative.

- 1) Equations in manuscripts are not numbered, and the equation in Page 26 is incomplete, please add.
- 2) There is a problem with the variables on Line 682,683 of page 24, please correct it.
- 3) The input sequence length of the multi-layer model needs to be fixed at 1234 amino acids. What is the basis for determining the input length? What is the distribution of sequence length in the database? In addition, for the sequences longer than the constraint length, will some important information be lost after interception?
- 4) Have the authors considered encoding each amino acid of the sequences as a vector with a length of 20? This could better reflect the position information of amino acids in the sequence.
- 5) In the section 'Introduction', it is mentioned 'to generate a priori knowledge on regulatory proteins, including kinase and phosphatase activity, we developed a scalable approach that determines sequence functionality using deep learning.' It will be nice if the authors can investigate interpretability of the 'prior knowledge'.
- 6) Since the model includes a CNN layer, an attention layer and a biLSTM layer, performing an ablation test provide more detail regarding the impact of each layer of the model.
- 7) The input dataset from the Pfam database is unbalanced, the authors should design an effective approach to the classification problem for the unbalanced dataset, and introduced appropriate performance evaluation metrics for the imbalanced classification study, such as the area under the precision-recall curve (AUPR) in the binary classification problem.

Reviewer #2 (Remarks to the Author):

The authors developed their neural network model, called Protein Family classification NETWORK (PF-NET), with four different layers. PF-NET can classify protein sequences into one of 996 protein families in Pfam. They demonstrated that PF-NET achieved sensitivities of 89% and 66%, respectively, for kinases and phosphatases by application of the selected threshold for all proteins of yeast proteome. Moreover, they indicated that they can augment protein families with novel members based on the higher predictive power of PF-NET than that of HMMER. They also presented their results of identification of new kinases and phosphatases in plants using PF-NET. They inferred regulatory networks in soybean using a dynamic Bayesian network approach, available in NetPhorce based on a time-course phosphoproteomics dataset which they measured, and on a list of kinase/phosphatase

proteins obtained using PF-NET (and other methods). They conclude that they were able to use the information extracted from PF-NET to produce an analytic pipeline that allows for the annotation and exploration of signaling pathways in non-model organisms.

Major:

Considering the goals of this study, we presume that the authors should clarify the impact of the novel knowledge obtained (solely) by PF-NET on their network inference. For instance, the authors state the newly predicted kinase by PF-NET: CEX1. How do such findings affect the results of their network inference when compared with the situation without those findings?

As the authors have described, “deep learning has been implemented to expand the coverage of the Pfam database and further annotate unknown proteins” using ProtCNN and ProtENN. In addition, other methods, such as models tested by Lee and Nguyen (<https://cs224d.stanford.edu/reports/LeeNguyen.pdf>), have been used for protein classification. Furthermore, deep neural “language” models such as ProteinBERT have been developed and have been applied to protein classification tasks. We presume that the authors might be able to apply this (or these) model(s) instead of PF-NET. Therefore, rather, they should compare the results obtained when using PF-NET with those obtained using a model such ProtCNN, instead of HMMER. Otherwise, we assume that developing a model particularly addressing identifying kinases and phosphatases, instead of 996 classes, based on a pretrained model might be a good way to obtain better classification results.

The authors should describe or explain, or at least mention those neural network models used for protein classification tasks, in addition to DeepBind, DeepSea, DanQ, and TBiNet, in the Introduction of the text.

According to Fig. 3C, 45 proteins were annotated to phosphatases solely by ortholog search. According to Fig. S3B, HMMER is rather effective for the case of *Sorghum bicolor* (sorghum). Based on the results presented in Fig. 5 and Fig. S3D, PF-NET is apparently more powerful for the case of *Oryza sativa* (rice) than other cases. It would be better if the author could state the unique strengths of the respective methods and the reasons for those strengths.

Minor:

According to the report by Lee and Nguyen, “it seems that more complex models (such as the bidirectional LSTM) are not required for high accuracy”, although they used 589, instead of 996, protein classes.

Page 14

Line 406: ... depending on the desired strincency. <- stringency (probably)

Please provide the full name for “PTPs”. Always spell out and define acronyms at first use.

Reviewer #3 (Remarks to the Author):

The author developed a neural network approach to annotate kinases and phosphatases in non-model species and show that their approach is complementary to conventional annotation using Hidden Markow models. This neural network is then use to annotated kinases and phosphatases in yeast to benchmark their predictions and subsequently

functionally annotated Arabidopsis and Soybean. The list of (newly) functionally annotated kinases and phosphatases in soybean was then used as a priory knowledge for inference of phosphorylation cascades in soybean after a time course of cold exposure.

Differential phosphosites were identified using a newly developed pipeline called NetPhorce. Finally, the authors modeled regulatory interactions between kinases and phosphatase and their potential substrates using the phosphoproteome data and functionally annotated kinases and phosphatases. Three kinase are identified from their network prediction, two of which were previously shown in Arabidopsis to be regulators of heat stress. As an encore the neural network was used to functionally annotate phosphatases in 4 crops species.

The neural network approach is very interesting and may have great potential for modelling signaling networks in (non-) model species. The manuscript is well written, and the modeling appears sound, but as a non-expert I'll mainly comment on the phosphoproteomics and signaling aspects of the manuscript.

Main comments

My main concern is that the use of NetPhorce to quantify label free phosphoproteomics data makes it virtually impossible to evaluate the data quality. Had NetPhorce been described before and benchmarked against other, more commonly used, data analysis software one could better judge this. As is, one has to assume the authors made all the right choices and set the appropriate thresholds to obtain a solid data set.

In particular, how did the author arise to a single p-value and q-value for a 10 point time course cold treatment? What threshold were chosen for differential phosphorylation? These points are of serious concern as there appears to be a large spread in the data points shown in figure 4 D, especially for the panels showing SRF6, TOI5 and TOT3. The authors may consider supplying more conventional data visualization for this type of phosphopeptide data, for example PCA analysis, volcano plots and/or heatmaps.

How do the authors explain that out of the 372 phosphopeptides that are quantified, 310 are classified as differential. Even with very robust signalling events, one would not expect more than 10-20 % of the quantified phosphopeptide to be differential.

Lines 670-675 The subdivision of the dataset into data that was not subjected to statistical analysis and data that was need further justification. If not enough data points are available, data should be filtered out rather than subcategorized as present/absent in my opinion

Lines 711-713 "A protein is a potential regulator of a target protein if and only if it exhibits a change in phosphorylation intensity at the same time (time lapse 0) or immediately prior (time lapse 1) to a
713 change in phosphorylation intensity of the target for at least 50% of the time points" This definition of a potential regulator of a target protein is flawed. If a potential regulator of a target protein exhibits a change in phosphorylation intensity at the same time one could equally argue these proteins share the same upstream kinase and no kinase substrate relationship exists. The definition therefor overestimates the number of regulator substrate relations

Minor comment

Line 150 biLSTM acronym not defined. Please write out in full

Line 724 formula lost it's parameters

We have addressed the concerns of the reviewers in the revised manuscript entitled "Functional annotation of proteins for signaling network inference in non-model species" submitted to *Nature Communications*. We thank the reviewers for their overall positive evaluation and critical comments to further improve our manuscript. Please find below detailed responses to the reviewers' comments.

Reviewer #1 (Remarks to the Author):

The authors presented a multi-layer neural network consisting of convolutional neural network (CNN) and bidirectional long short term memory (biLSTM) to construct feature representations and classify proteins into 996 protein families for protein function annotation. Moreover, a network inference method with a dynamic Bayesian network was developed to predict signaling cascades. The authors conducted adequate experiments on six species including *Arabidopsis thaliana*, soybean, wheat, maize, rice, and sorghum, and the results demonstrated that the proposed method is highly generalizable and applicable in protein function annotation. However, taking care of below points, will make the paper more informative.

- 1) Equations in manuscripts are not numbered, and the equation in Page 26 is incomplete, please add.

REPLY: We have now numbered all the equations in the revised version of the manuscript.

- 2) There is a problem with the variables on Line 682,683 of page 24, please correct it.

REPLY: Thank you for pointing out this discrepancy with the variables. An error occurred when uploading the PDF to the submission platform. We have corrected the issue.

- 3) The input sequence length of the multi-layer model needs to be fixed at 1234 amino acids. What is the basis for determining the input length? What is the distribution of sequence length in the database? In addition, for the sequences longer than the constraint length, will some important information be lost after interception?

REPLY: As previously demonstrated in the study conducted by Lopez-del Rio et al. (2020) in *Nature*, sequence padding significantly impacts model performance. In order to mitigate this effect, we established a common length of 1234 amino acids for all proteins. For shorter sequences, we padded them, and for longer sequences, we split them into 1234 pieces, while retaining maximum information. To determine this common length, we plotted the distribution of the input lengths of the first batch of sequences in a histogram with 50 bins (representing approximately 1 million sequences) (Fig. 1A). We found that the majority of the sequences (98.4%) were distributed within the first 6 bins (< 1234 amino acids) and only 1.6% of the sequences required splitting. To substantiate our choice of common length, we performed an additional analysis to identify the optimal number of bins using the "auto" option (Maximum of the Sturges and Freedman Diaconis Estimator estimators) (Fig. 1B) and, subsequently, identified the inflection-point or "knee" in the cumulative plot using a maximum curvature approach (Fig. 1C-D) (Satopaa et al, 2011). The inflection point indicates at which point the addition of sequences per bin is increasing in a linear fashion. At this point, the relative costs to increase the amino acid length and thus increase the sequence padding is no longer worth the added sequences

and reduced splits. Our analysis identified the inflection point at 1210 amino acids, which is within a 5% confidence interval of our chosen common length (i.e. 1234).

Fig. 1 – (A, B) Histogram of the sequence lengths within the input dataset of PF-NET with 50 bins (A) or auto identified number of bins (B). (C) The cumulative number of sequences for each bin. (D) The normalized curve, where x and y values are normalized between 0 and 1, upon fitting a polynomial curve to identify the knee point. The dotted red line represents the chosen common length to which the input sequences are padded or split and the dotted blue line represents the knee/elbow point.

- 4) Have the authors considered encoding each amino acid of the sequences as a vector with a length of 20? This could better reflect the position information of amino acids in the sequence.

REPLY: We employed a binary encoding strategy wherein each amino acid is represented by a binary code consisting of 5 bits. Binary encoding offers the benefit of reducing the number of input columns required (5 columns for binary encoding versus 21 columns for one-hot encoding) thereby decreasing the dimensionality of the dataset and making it more manageable. Our high performance computing system (Table 1) with its current specifications allowed for sufficient memory and the runtime was within weeks using binary encoding. If we were to use a one-hot encoding scheme, it would increase the needed memory. Notably, since each amino acid is represented by a unique binary code, the positional information of amino acids within the sequence is preserved. Given that we could retain positional information and manage memory and runtime, we choose to use binary encoding with 5 bits.

We have updated the methods section (Lines 513 to 516) to include this information: “Binary encoding has the advantage that the number of input columns (5 columns for

binary encoding versus 21 columns for one-hot encoding) and thus dimensionality of the dataset requires less memory. The binary encoding scheme requires 128GB RAM to load.”

Table 1 – Machine Learning Specifications.

CPU Specifications	GPU (x10)
Intel Xeon Gold 6258R x2	NVIDIA A100 Tensor Core GPU
Total Cores: 28	GPU Memory 80GB: HBM2e
Total Threads: 56	FP64: 9.7 TFLOP
Max Turbo Frequency: 4.0 GHz	FP64 Tensor Core: 19.5 TFLOPS
Processor Base Frequency: 2.70 GHz	Tensor Float 32 (TF32): 156 TFLOPS
Cache: 38.5 MB	BFLOAT16 Tensor Core: 312 TFLOPS
TDP (Thermal Design Power): 205W	FP16 Tensor Core: 312 TFLOPS
	INT8 Tensor Core: 624 TOPS
	Memory size 80 GB
	Memory bus width 5120 bits
	Peak memory bandwidth Up to 1.94 TB/s
Memory	
1.5 TBs DDR4 RDIMMs (64GB x 24)	
Memory rated at 3200 MT/s	

- 5) In the section ‘Introduction’, it is mentioned ‘to generate a priori knowledge on regulatory proteins, including kinase and phosphatase activity, we developed a scalable approach that determines sequence functionality using deep learning.’ It will be nice if the authors can investigate interpretability of the ‘prior knowledge’.

REPLY: Our interpretation of the reviewer’s question is that they are inquiring about the importance of prior knowledge, specifically a list of kinases and phosphatases, in inferring regulatory networks. The inclusion of prior knowledge has several advantages, with the primary one being a significant improvement in the accuracy of the network inference methods. Without prior knowledge of the upstream regulators, well-established network inference methods yield predictions that were not better than random guessing (Huynh-Thu et al., 2010). Other advantages include a reduced number of computations, a smaller subset of regulators need to be considered that makes the computation less complex, and the directionality from regulator to target is automatically correct. All these benefits result in increased network inference accuracy. Therefore, identifying the full extent of the plethora of upstream regulators is critical to fully map the regulatory modules of a protein’s phosphorylation events and the complexity of signaling networks (Cargnello et al., 2011; Picton et al., 1982). During our network inference, newly predicted kinases or phosphatase were considered as potential upstream regulators as they may be biologically important kinases/phosphatases and thus relevant for further in-depth characterization studies. The omission or neglect of upstream regulators (known and novel) would lead to incomplete predictions and systemic errors. Thus, integrating the novel knowledge from PF-NET for network inference reduces uncertainties and enhances overall accuracy to unravel the dynamics of a signaling cascade and observing the progression of phosphorylation events.

To highlight the impact of prior knowledge on network inference, we added a paragraph in the introduction and the discussion. In the introduction (Lines 92 to 94): “Specifically, the prior knowledge of regulatory proteins significantly improves network inference accuracy. Without prior knowledge of the upstream regulators, well-established network

inference methods yield predictions that were not better than random guessing (Huynh-Thu et al. 2010).”

In the discussion (Lines 414 to 421): “For example, CEX1 was predicted to function as a kinase solely by PF-NET. Identifying new kinases/phosphatases impacts downstream analysis as the inclusivity of regulators reduces uncertainties and increases overall accuracy for network inference. Omitting or not considering novel predicted regulators would lead to incomplete predictions and systemic errors. In addition, many target proteins can be (de)phosphorylated by multiple upstream kinases/phosphatases, resulting in a complex regulatory module for every target protein. Identifying the full extent of this plethora of upstream regulators is thus critical to fully map the regulatory modules of a protein’s phosphorylation events and, concordantly, the complexity of signaling cascades.”.

- 6) Since the model includes a CNN layer, an attention layer and a biLSTM layer, performing an ablation test provide more detail regarding the impact of each layer of the model.

REPLY: We have now included an ablation test to shed light on the impact of each layer on the model’s performance. To this end, we generated three models wherein for each model a layer was removed. Since the attention layer attenuates the features from the previous hidden layer, we removed both CNN and attention layer to assess the impact of the CNN layer. To compare the performance of these models with our full original model, we trained the models using the same training scheme and with the exact same training, validation, and test data on the first batch combination of sequences (representing approximately 3.6 million sequences). Upon removal of the biLSTM layer or both the CNN and attention layers, the model performance dropped across all metrics (Fig. 2A). The layer with the highest impact was the biLSTM, which indicates that the biLSTM layer is crucial for a high model performance. While the removal of the CNN and attention layer decreased the network performance, the removal of only the attention layer did not result in a decreased but an increased performance (Fig 2A). In contrast, when predicting motifs within DNA sequences instead of protein sequences, the addition of a attention layer actually increases performance, as was shown in TBiNet (CNN-attention-biLSTM), which improved upon DanQ (CNN-biLSTM) (Park et al., 2020). Overall, these results show that the CNN and biLSTM are crucial layers to predict protein function.

To ensure that the model without attention layer is not outperforming our original neural network, we fully trained this model on the five sequence batch combinations and compared performance. The performance of both models was almost identical (Fig 2B,C). Understanding the influence of each layer is crucial, and an ablation study allowed us to investigate the causation of the individual parts to the overall system.

To highlight the impact of each layer on the model, we added a paragraph on the ablation study in the results section (Line 163 to 166) : “To gain a better understanding of the contribution of the individual layers, we removed each layer and evaluated the performance of these new models. This ablation study showed that especially the CNN and biLSTM are crucial for PF-NET’s performance (Supplementary Fig. S2).”.

Fig. 2 (included in the manuscript as Supplemental Fig. S2) – Neural network performance upon an ablation study of each of its three layers. (A) The performance of PF-NET (original model) and three models where the biLSTM, the attention layer, and both the CNN and attention later are removed. The performance is given of the validation dataset after training with one batch combination of sequences (~ 3.6 million sequences) (B) The performance of PF-NET and the same model without attention layer after training on the full five batch combinations. (C) The overall performance of the validation dataset of the models in B in terms of area under the precision curve (AUPR) , f1-score, recall, and precision during training.

7) The input dataset from the Pfam database is unbalanced, the authors should design an effective approach to the classification problem for the unbalanced dataset, and introduced appropriate performance evaluation metrics for the imbalanced classification study, such as the area under the precision-recall curve (AUPR) in the binary classification problem.

REPLY: As requested by the reviewer, we have computed the AUPRC for all the classes, and the macro and weighted average. For the weighted average, the scoring metrics for each class were calculated, weighted by the number of true instances for each label, and

averaged to account for class imbalance. We have added the macro and weighted averages in main Figure 1 B, the AUPRC for the kinase and phosphatase families in main Figure 1 E, the AUPRC per class in the Supplementary Data S2, and the AUPRC during training in Supplementary Figure S1. Overall, the AUPRC was between 0.898 and 0.999 for the kinases and phosphatases, indicating that PF-NET can accurately predict these classes.

As an effective approach to the classification problem for the unbalanced dataset, we deployed cost-sensitive approaches rather than sampling methods. Undersampling would have reduced the data points for the majority classes considerably leading to incorrect classification, while oversampling of the minority class adds additional noise to the dataset. Thus, we choose to deploy cost-sensitive approaches rather than sampling methods. Specifically, our cost sensitive approach is twofold; (i) the misclassification of the minority class penalizes, and (ii) our class weights are inversely proportional to the class sizes in the dataset.

Fig. 3 (Included in manuscript as Supplementary Fig. S1) – Classification performance of PF-NET. The overall performance of PF-NET in terms of loss, accuracy, precision, f1-score, recall, and area under the precision curve (AUPRC) during training and validation.

Reviewer #2 (Remarks to the Author):

The authors developed their neural network model, called Protein Family classification NETWORK (PF-NET), with four different layers. PF-NET can classify protein sequences into one of 996 protein families in Pfam. They demonstrated that PF-NET achieved sensitivities of 89% and 66%, respectively, for kinases and phosphatases by application of the selected threshold for all proteins of yeast proteome. Moreover, they indicated that they can augment protein families with novel members based on the higher predictive power of PF-NET than that of HMMER. They also presented their results of identification of new kinases and phosphatases in plants using PF-NET. They inferred regulatory networks in soybean using a dynamic Bayesian network approach, available in NetPhorce based on a time-course phosphoproteomics dataset which they measured, and on a list of kinase/phosphatase proteins obtained using PF-NET (and other methods). They conclude that they were able to use the information extracted from PF-NET to produce an analytic pipeline that allows for the annotation and exploration of signaling pathways in non-model organisms.

Major:

Considering the goals of this study, we presume that the authors should clarify the impact of the novel knowledge obtained (solely) by PF-NET on their network inference. For instance, the authors state the newly predicted kinase by PF-NET: CEX1. How do such findings affect the results of their network inference when compared with the situation without those findings?

REPLY: The inclusion of prior knowledge has several advantages, with the primary one being a significant improvement in the accuracy of the network inference methods. Without prior knowledge of the upstream regulators, well-established network inference methods yield predictions that were not better than random guessing (Huynh-Thu et al., 2010). Other advantages include a reduced number of computations, a smaller subset of regulators need to be considered that makes the computation less complex, and the directionality from regulator to target is automatically correct. All these benefits result in an increased accuracy of network inference.

Many target proteins can be (de)phosphorylated by multiple upstream kinases/phosphatases, resulting in a complex regulatory module for every target protein (Cargnello et al., 2011; Picton et al., 1982). Therefore, identifying the full extent of the plethora of upstream regulators is critical to fully map the regulatory modules of a protein's phosphorylation events and the complexity of signaling networks (Cargnello et al., 2011; Picton et al., 1982). During our network inference, newly predicted kinases or phosphatase were considered as potential upstream regulators as they may be biologically important kinases/phosphatases and thus relevant for further in-depth characterization studies. The omission or neglect of upstream regulators (known and novel) would lead to incomplete predictions and systemic errors. Thus, integrating the novel knowledge from PF-NET for network inference reduces uncertainties and enhances overall accuracy to unravel the dynamics of a signaling cascade and observing the progression of phosphorylation events.

To highlight the impact of prior knowledge on network inference, we added a paragraph in the introduction and the discussion. In the introduction (Lines 92 to 94): "Specifically, the prior knowledge of regulatory proteins significantly improves network inference accuracy. Without prior knowledge of the upstream regulators, well-established network inference methods yield predictions that were not better than random guessing (Huynh-Thu et al. 2010)."

In the discussion (Lines 414 to 421): “For example, CEX1 was predicted to function as a kinase solely by PF-NET. Identifying new kinases/phosphatases impacts downstream analysis as the inclusivity of regulators reduces uncertainties and increases overall accuracy for network inference. Omitting or not considering novel predicted regulators would lead to incomplete predictions and systemic errors. In addition, many target proteins can be (de)phosphorylated by multiple upstream kinases/phosphatases, resulting in a complex regulatory module for every target protein. Identifying the full extent of this plethora of upstream regulators is thus critical to fully map the regulatory modules of a protein’s phosphorylation events and, concordantly, the complexity of signaling cascades.”.

As the authors have described, “deep learning has been implemented to expand the coverage of the Pfam database and further annotate unknown proteins” using ProtCNN and ProtENN. In addition, other methods, such as models tested by Lee and Nguyen (<https://cs224d.stanford.edu/reports/LeeNguyen.pdf>), have been used for protein classification. Furthermore, deep neural “language” models such as ProteinBERT have been developed and have been applied to protein classification tasks. We presume that the authors might be able to apply this (or these) model(s) instead of PF-NET. Therefore, rather, they should compare the results obtained when using PF-NET with those obtained using a model such ProtCNN, instead of HMMER. Otherwise, we assume that developing a model particularly addressing identifying kinases and phosphatases, instead of 996 classes, based on a pretrained model might be a good way to obtain better classification results.

REPLY: To substantiate the application of PF-NET rather than existing models, we compared the performance of PF-NET to a deep learning model. We selected ProtCNN to compare PF-NET’s performance instead of the models from Lee and Nguyen et al and ProteinBert for the following reasons. The models trained by Lee and Nguyen et al were trained on 550,960 sequences, while we use more than 7 million sequences as input. In addition they used 589 families, for which the exact families are not presented, nor are the models publicly available to test and compare performance. Since ProteinBERT performs Gene Ontology (GO) annotation prediction, which covers molecular functions, biological processes and subcellular locations (Brandes et al 2022), but not Pfam annotations, this neural network is complementary to PF-NET.

We performed protein classification with ProtCNN using the provided interactive notebook that runs inference of the fully trained ProtCNN model. Similarly as for the performance comparison with HMMER, we used the yeast proteome as a benchmark dataset. ProtCNN’s preprocessing first pads the sequences to the maximum length in the provided dataset, which equals to 4053 AA for the yeast proteome. To ensure that a difference in sequence padding would not influence our comparisons, we ran the predictions with two maximum sequence lengths. On the one hand, we used the maximum sequence length of 4053 AA, and, on the other hand, we used our set sequence length of 1234 AA to predict protein function. Both padding schemes rendered the same results. Upon evaluating ProtCNN’s predictions of the yeast proteome, we found that ProtCNN was unable to retrieve most kinases and phosphatases and both PF-NET and HMMER greatly outperformed ProtCNN (Fig. 4). ProtCNN predicted only one kinase and five phosphatases (Fig. 4). We reasoned that since ProtCNN is trained on protein domains rather than full length sequences, the poor predictions of ProtCNN are primarily due to the input sequences. For example, when using solely the kinase domain sequence of the yeast kinase

PSK1 as input, the correct class (PF00069) was retrieved by ProtCNN, while running ProtCNN using the full length sequence, functionally predicted PSK1 as a Myelin transcription factor 1 (PF08474). Because positional information on the domain sequence is generally unknown for non-model species and our objective is to functionally annotate proteins from non-model species efficiently, we choose to use the full length protein sequences as input. In addition, using the full length sequence has the benefit of taking into account the entire protein sequence, which leads to the prediction of novel protein kinases, such as CEX1, which can be missed by domain-specific prediction methods.

We added the comparison of the yeast proteome predictions for PF-NET, HMMER, and ProtCNN to the results section (Lines 222 to 225): “We also compared PF-NET’s performance to ProtCNN, a protein classification neural network. However, as ProtCNN is trained on protein domain sequences rather than the full protein sequence, a poor performance was retrieved (Supplementary Fig. S4).”

Fig. 4 (Included in the manuscript as Supplemental Fig. S4) – Functional predictions of kinases and phosphatases in *Saccharomyces cerevisiae*. (A-B) Commonly identified kinases (A) and phosphatases (B) by PF-NET, HMMER search, ProtCNN, and the ground truth. (E) The overall performance of PF-NET and HMMER in terms of sensitivity and false discovery rate (FDR).

The authors should describe or explain, or at least mention those neural network models used for protein classification tasks, in addition to DeepBind, DeepSea, DanQ, and TBiNet, in the Introduction of the text.

REPLY: We added a short description of ProtCNN and ProteinBERT to introduction as requested by the reviewer. Line 83 to 89: “Existing machine learning approaches that predict protein function do so by predicting gene ontology or protein structure associated with the amino acid sequence. For example, ProteinBERT is a deep-learning language model that inputs GO annotations and protein sequences and shows state-of-the-art performance on predicting protein

structure, post-translational modifications, and biophysical properties. A recent study showed the potential of machine learning models to complement existing approaches for protein function prediction tools. Specifically, ProtCNN, a neural network trained on protein domain sequences, accurately annotated protein domains and improved and expanded on current proteins annotations.”.

According to Fig. 3C, 45 proteins were annotated to phosphatases solely by ortholog search. According to Fig. S3B, HMMER is rather effective for the case of Sorghum bicolor (sorghum). Based on the results presented in Fig. 5 and Fig. S3D, PF-NET is apparently more powerful for the case of *Oryza sativa* (rice) than other cases. It would be better if the author could state the unique strengths of the respective methods and the reasons for those strengths.

REPLY: When investigating the strengths of the respective methods applied to diverse crop species, we were able to elaborate on our predictions in crops and highlight strengths of our method. To shed light on the clear knowledge gap of each methods’ strength, we compared phylogenetic relations and sequence similarities among the phosphatases.

There is a prevailing notion that functional information can be transferred from one species to another. However, the connection between biological function and orthology can decrease as evolution continues, duplications occur, subfunctionalization happens, and protein function changes (Koonin et al., 2005). *Arabidopsis thaliana* and *Glycine max* are both dicots, while the other four crops (wheat, maize, rice, and sorghum) are monocots. The first bipartition in the phylogenetic species tree in Fig. 5C in the manuscript separates dicots and monocots. Since *Arabidopsis* and soybean are thus evolutionary the least divergent (Fig. 5C in manuscript), the orthologs predict the highest number of kinases and phosphatases for soybean.

To understand why HMMER or PF-NET would identify an increasing number of phosphatases, we tested if the sequence similarity might be a probable cause. It has been shown that the error rate of sequence alignment methods increases when sequence similarity decreases (Bileschi et al., 2022). HMMER is such a method that aligns sequences to a HMM profile. To test whether sequence similarity might be correlated with a higher number of predicted sequences for HMMER compared to PF-NET, we performed a multiple sequence alignment of the predicted phosphatases of each species using Clustal Omega, which is suitable for medium-large alignments (Sievers et al., 2011). To easily visualize the differences in sequence similarity between the species, we plotted the resulting percent identity matrices (Fig. 5). The maize phosphatases appear to show a higher similarity compared to the other species. Concurrently, HMMER is predicting a higher number of phosphatases in maize compared to PF-NET. For species with lower sequence similarity, including rice and wheat, PF-NET is predicting a higher number of sequences. The higher sequence similarity for phosphatases in maize might be because the maize genome is repleted with chromosomal duplications and repetitive DNA (Gaut et al., 2000).

We added these results to the manuscript (Line 352 to 354): “Species for which PF-NET/HMMER would identify an increasing number of phosphatases, showed a lower/higher sequence similarity, respectively (Supplementary Fig. S7).”.

Fig. 5 (Included in the manuscript as Supplementary Fig. S7) – Sequence percent identity matrices for phosphatases from Arabidopsis, soybean, sorghum, maize, rice, and wheat. The axes contain all the predicted phosphatases clustered according to hierarchical clustering. Color scale ranges from white (minimum value) to orange (maximum value).

Minor:

According to the report by Lee and Nguyen, “it seems that more complex models (such as the bidirectional LSTM) are not required for high accuracy”, although they used 589, instead of 996, protein classes.

REPLY: We have now included an ablation test and were able to shed light on the impact of each of the layers on the performance. To this end, we generated three models wherein for each model a layer was removed. Since the attention layer attenuates the features from the previous hidden layer, we removed both CNN and attention layer to assess the impact of the CNN layer. To compare the performance of these models with our full original model, we trained the models using the same training scheme and with the exact same training, validation, and test data on the first batch combination of sequences (representing approximately 3.6 million sequences). Upon removal of the biLSTM layer or both the CNN and attention layers, the model performance dropped across all metrics (Fig. 2A). The layer with the highest impact was the biLSTM, which indicates that the biLSTM layer is crucial for a high model performance. While the removal of the CNN and attention layer decreased the network performance, the removal of only the attention layer did not result in a decreased but an increased performance (Fig 2A). In contrast, when

predicting motifs within DNA sequences instead of protein sequences, the addition of a attention layer actually increases performance, as was shown in TBiNet (CNN-attention-biLSTM), which improved upon DanQ (CNN-biLSTM) (Park et al., 2020). Overall, these results show that the CNN and biLSTM are crucial layers to predict protein function.

To ensure that the model without attention layer is not outperforming our original neural network, we fully trained this model on the five sequence batch combinations and compared performance. The performance of both models was almost identical (Fig 2B,C). Understanding the influence of each layer is crucial, and an ablation study allowed us to investigate the causation of the individual parts to the overall system.

To highlight the impact of each layer on the model, we added a paragraph on the ablation study in the results section (Line 163 to 166) : “To gain a better understanding of the contribution of the individual layers, we removed each layer and evaluated the performance of these new models. This ablation study showed that especially the CNN and biLSTM are crucial for PF-NET’s performance (Supplementary Fig. S2).”.

Fig. 2 (included in the manuscript as Supplemental Fig. S2) – Neural network performance upon an ablation study of each of its three layers. (A) The performance of PF-NET (original model) and three models where the biLSTM, the attention layer, and both the CNN and attention later are removed. The performance is given of the validation dataset after training with one batch combination of sequences (~ 3.6 million sequences) (B) The performance of PF-NET and the same model without attention layer after training on the full five batch combinations. (C) The overall performance of the validation dataset of the models in B in terms of area under the precision curve (AUPR) , f1-score, recall, and precision during training.

Page 14 - Line 406: ... depending on the desired strincency. <- stringency (probably)

REPLY: We have corrected *strincency* to *stringency*

Please provide the full name for “PTPs”. Always spell out and define acronyms at first use.

REPLY: We spelled out PTPs at first use in the first result section: “Thus, to identify the undescribed phosphatases in soybean, we performed a HMMER search ³ with HMMs for Ser/Thr phosphatases (STs), dual-specificity phosphatases (DSPs), protein phosphatases 2C (PP2Cs), and protein tyrosine phosphatases (PTPs).”

Reviewer #3 (Remarks to the Author):

The author developed a neural network approach to annotate kinases and phosphatases in non-model species and show that their approach is complementary to conventional annotation using Hidden Markow models. This neural network is then use to annotated kinases and phosphatases in yeast to benchmark their predictions and subsequently functionally annotated Arabidopsis and Soybean. The list of (newly) functionally annotated kinases and phosphatases in soybean was then used as a priory knowledge for inference of phosphorylation cascades in soybean after a time course of cold exposure.

Differential phosphosites were identified using a newly developed pipeline called NetPhorce. Finally, the authors modeled regulatory interactions between kinases and phosphatase and their potential substrates using the phosphoproteome data and functionally annotated kinases and phosphatases. Three kinase are identified from their network prediction, two of which were previously shown in Arabidopsis to be regulators of heat stress. As an encore the neural network was used to functionally annotate phosphatases in 4 crops species.

The neural network approach is very interesting and may have great potential for modelling signaling networks in (non-) model species. The manuscript is well written, and the modeling appears sound, but as a non-expert I'll mainly comment on the phosphoproteomics and signaling aspects of the manuscript.

Main comments

My main concern is that the use of NetPhorce to quantify label free phosphoproteomics data makes it virtually impossible to evaluate the data quality. Had NetPhorce been described before and benchmarked against other, more commonly used, data analysis software one could better judge this. As is, one has to assume the authors made all the right choices and set the appropriate thresholds to obtain a solid data set.

REPLY: We now benchmarked the data analysis method from NetPhorce to an existing data analysis software. Specifically, we used the well-established Perseus software platform (Tyanova et al., 2016). In Perseus, we filtered the rows and removed potential contaminants and reverse peptides detected by MaxQuant, which resulted in 8081 phosphopeptides, similar as in NetPhorce. We further filtered the rows based on at least three valid values out of four replicates per sample, which is the same criteria we applied in NetPhorce and a common approach. The filtering resulted in 361 phosphopeptides, which were used for statistics. A two-way ANOVA within Perseus resulted in 253 (~ 70.1%) phosphopeptides significant for treatment. In NetPhorce, we fit a model that in addition to the treatment and time factor, includes the replicates as a random factor. Since MaxQuant searches were performed within replicates with both control and cold treatment groups, including replicates as a random factor leads to more powerful/sensitive statistics. Thus, NetPhorce detects a majority of significant phosphopeptides (~ 85.9%) compared to Perseus. Overall, we found that the data analysis in NetPhorce and Perseus are comparable.

Table 1 (Included in the manuscript as Supplementary Table S1) – Comparison between NetPhorce and Perseus at various intermediate steps of the data analysis of the soybean cold phosphoproteomics.

	Perseus	NetPhorce
Potential contaminant and Reverse (phosphopeptides)	8081	8081
Valid values filtering (phosphopeptides)	361	361
Valid values filtering (proteins)	320	320
Statistics (phosphopeptides)	253 (70.1%)	310 (85.9%)
Absence/Presence	NA	11

To substantiate the use of NetPhorce for quantifying phosphoproteomics, we added a paragraph on the benchmarking NetPhorce with Persues to the methods section (Lines 756 to 757): “All intermediate steps of our pipeline were benchmarked against the Perseus software platform (Tyanova et al., 2016) and showed the same results (Supplementary Table S1).”.

Please note that filtering of the phosphosites for localization probability was done prior to data analysis in NetPhorce/Perseus. Since we processed the MaxQuant searches within replicates, four distinct localization probabilities (1 per Phospho(STY)file) were retrieved. Thus, to select phosphosites that were reliably identified and quantified in all replicates (4/4 being ≥ 0.75), we applied a ≥ 0.75 filter in each and every replicate phospho-dataframe. Finally, the merged localization probability in the final table reports the average of all localization probabilities for that particular phosphosite in all analyzed replicates.

In particular, how did the author arise to a single p-value and q-value for a 10 point time course cold treatment? What threshold were chosen for differential phosphorylation? These points are of serious concern as there appears to be a large spread in the data points shown in figure 4 D, especially for the panels showing SRF6, TOI5 and TOT3. The authors may consider supplying more conventional data visualization for this type of phosphopeptide data, for example PCA analysis, volcano plots and/or heatmaps.

REPLY: To employ statistics that can evaluate more complex experimental designs, NetPhorce implements a linear mixed model where the treatment, time, and their interaction term were included as fixed effects and the replicate variable as a random effect. The mixed model analysis outputs three p-values for each of the two fixed variables and their interaction term (time:treatment). As our objective was to identify factors key for the cold response in soybean seedlings, we selected the significantly differentially phosphorylated peptides upon cold (i.e. a p-value of the treatment variable < 0.05). To further account for multiple testing we used the distribution of the p-values to calculate q-values with the R package qvalues (Storey et al., 2021). All peptides with a q value smaller than 0.05 were considered significant differentially phosphorylated.

To further clarify the selection criteria for differential phosphorylated, we added the following in the methods section (Line 749 to 753): “As our objective was to identify factors key for the cold response in soybean seedlings, we selected the significantly differentially phosphorylated peptides upon cold (i.e. a p-value of the condition variable < 0.05). Generally, NetPhorce will select the p-value of the condition variable unless the input data does not contain a condition variable, in that case, NetPhorce will consider the p-value of the time variable.”.

As requested by the reviewer, we also visualized the phosphopeptides in a heatmap and volcano plot (Fig. 6), which were included in the manuscript as Supplementary Fig. S5.

Fig. 6 (included in the manuscript as Supplementary Fig. S5) – Significant differentially phosphorylated phosphopeptides upon cold in soybean. (A) Heatmap of the log₂ fold change upon cold across the entire time course. (B) A volcano plot that displays the maximum fold change of the time course in respect to the -log₁₀ qvalue.

How do the authors explain that out of the 372 phosphopeptides that are quantified, 310 are classified as differential. Even with very robust signalling events, one would not expect more than 10-20 % of the quantified phosphopeptide to be differential.

REPLY: When analyzed with the phosphoproteomics time course with a two-way ANOVA in Perseus, we classified a total of 253 phosphopeptides as differential for treatment, which equals to ~ 70.1%. We classified 310 (~ 85.9%) phosphopeptides as differential with NetPhorce, which is a larger percentage due to the implementation of a more powerful/sensitive model that includes the replicates as a random variable. However, as pointed out by the reviewer both analyses render a high number of significant differential phosphopeptides. The high percentage of differential phosphopeptides is the result of two causes. (1) The large number of time points (11 time points) together with four replicates and two conditions each, gives a total sample size of 88, which results in a high statistical power. To showcase this high statistical power, we performed a power analysis to detect differences between the two treatments considering the 11 timepoints and 4 replicates. Depending on the differences between the two means that we aim to detect, a different power is achieved. To detect difference of 1.5 a two means comparison has a power of 90.28%, while a difference of two is even detected with 99.14% chance (Table 2-3). (2) Because the MaxQuant searches were performed within replicates and as a result of the large sample size, a low but highly confident number of phosphopeptides (361/8081 ~4.4%) with few or no missing values was retained for statistical analysis. This stringent selection of high confidence phosphopeptides and the high statistical power resulted in a large number of statistically differential phosphopeptides.

Table 2 – Input parameters for power calculations in JMP Pro 0.15 for testing that the means are different across 2 samples (in our case cold and control conditions). Standard deviation and sample size are calculated as the average of the standard deviations and sample sizes of each phosphopeptide in our phosphoproteomics.

Parameter	Value
Alpha (significance level)	0.05
Standard deviation (average SD of the phosphopeptides)	1.4
Sample size per treatment (average, taking missing values into consideration)	39

Table 3 – Outcomes of the power analyses for three set differences between the means to detect..

Differences to detect	Power
2	0.9914
1.5	0.9028
1	0.5841

Lines 670-675 The subdivision of the dataset into data that was not subjected to statistical analysis and data that was need further justification. If not enough data points are available, data should be filtered out rather than subcategorized as present/absent in my opinion

REPLY: We agree with the reviewer that when not enough data points are available, data should be filtered out. Indeed, our pipeline does filter out phosphopeptides with insufficient data points. However, similar to previous analyses (Vu et al., 2021; Nikonorova et al., 2018; Smith et al., 2020; Nikonorova et al., 2021) we acknowledged important aspects of missingness and kept a

specific case of missingness, which is biologically relevant and labeled here as present/absent. Generally, two types of missingness can be found within (phospho)proteomics: (1) missingness completely at random (MCAR), and (2) missingness not at random (MNAR) (Karpievitch et al., 2012). MCAR occurs due to technical or instrumentation defects, such as poor ionization, other peptides competing for charge, and enzymatic modifications. The missingness of such peptides in a sample is unrelated to its abundance. On the other hand, MNAR is related to the peptide's abundance and occurs when a peptide abundance falls below the instrument detection limit or a peptide is simply not present. Phosphopeptides that are very lowly abundant or not present in one condition versus abundant in another condition (e.g. cold versus control conditions) are biologically relevant as they can for example be part of a pathway that is only activated upon cold (Vu et al., 2021; Nikonorova et al., 2018; Smith et al., 2020; Nikonorova et al., 2021). In our approach, we aimed to only include MNAR and discard MCAR, hence including the biologically relevant peptides but excluding the peptides with too few data points. To distinguish between MNAR and MCAR, we set up stringent criteria and only detect MNAR and thus absence of the phosphopeptides in a sample when the phosphopeptides is not detected in three or four out of the four replicates. A phosphopeptides needs to be classified as MNAR or have sufficient valid values (i.e. detected in three or four out of the four replicates) for all samples. When a peptide has in even one sample two out of the four valid values, the peptide is classified as MNAR for that sample, considered as insufficient data points and is filtered out as suggested by the reviewer.

We elaborated on further on the missingness briefly in the results and more elaborate in the methods.

Results lines 291 to 293: "The 8081 detected phosphosites were used as input for our pipeline, which first performs several quality control steps, including the removal of phosphosites with insufficient datapoints (see Methods)."

Results lines 299 to 302: "To include very lowly abundant or absent phosphosites that fall below the detection limit and thus to overcome the intrinsic detection limitation of phosphoproteomics, we included those phosphosites that were undetected in the majority or in all replicates of a sample and considered their phosphorylation absent."

Methods lines 723 to 733: "Generally, two types of missingness can be found within (phospho)proteomics: (1) missingness completely at random (MCAR), and (2) missingness not at random (MNAR) (Karpievitch et al., 2012). MCAR occurs due to technical or instrumentation defects, such as poor ionization, other peptides competing for charge, and enzymatic modifications, and is unrelated to its abundance. On the other hand, MNAR is related to the peptide's abundance and occurs when a peptide abundance falls below the instrument detection limit or a peptide is simply not present. Phosphopeptides that are very lowly abundant or not present in one condition versus abundant in another condition (e.g. cold versus control conditions) are biologically relevant as they can for example be part of a pathway that is only activated upon cold. In our approach, we included MNAR and discarded MCAR, hence including the biologically relevant peptides but excluding the peptides with too few data points."

Lines 711-713 "A protein is a potential regulator of a target protein if and only if it exhibits a change in phosphorylation intensity at the same time (time lapse 0) or immediately prior (time lapse 1) to a change in phosphorylation intensity of the target for at least 50% of the time points"

This definition of a potential regulator of a target protein is flawed. If a potential regulator of a target protein exhibits a change in phosphorylation intensity at the same time one could equally argue these proteins share the same upstream kinase and no kinase substrate relationship exists. The definition therefor overestimates the number of regulator substrate relations

REPLY: We agree with the reviewer that this definition can be an overestimation of the number of regulator substrate relations. However, this depends on the sparsity of the time points in the dataset. A phosphorylation within the same time point could occur when the experimental design has longer time steps, for example for daily sampling, while an experimental design with shorter time steps will include phosphorylation with a time lag (Zou et al., 2005). To identify biologically relevant time steps for simultaneous or antecedent phosphorylation changes, we searched for publications that studied the phosphorylation time lag between potential regulators and their substrates. In mammalian cells, the phosphorylation of a downstream kinase upon stimulation of the receptor with growth factors ranges between 26.6 and 200 seconds (Blazek et al., 2015). In plants, phosphorylation of NPH3 at S744 was detected within 30 s after blue light activation of its upstream kinase PHOT1 and maintained over the 2 h irradiation period (Sullivan et al., 2021). In another example in plants, BIK1 is phosphorylated by flg22 within the first minutes upon stimulation. The authors suggest that BIK1 functions downstream of FLS2/BAK1 complex formation and phosphorylation, which is induced by flg22 (Schulze et al., 2010; Smith et al., 2013). Given our time course (every 6 minutes for one hour) and the fast nature of phosphorylation cascades, we selected time lapse 0 and 1 as we assumed that the regulation may happen within and between time points. To validate our conclusions, we compared the network topology when only including time lapse 1, where kinases/phosphatases with earlier changes in phosphorylation are assigned as the potential regulators of targets (Fig. 7, included in the manuscript as supplemental Fig. S8). In this reduced network, TOI5 still showed the highest centrality and rewiring upon cold (Table 3). TOT3 became the fourth most central node instead of the second most central, but still showed the third highest rewiring upon cold.

Table 3 (included in the manuscript as Supplementary Fig. S8) – The betweenness centrality and rewiring value calculated in Cytoscape for each of the upstream nodes from the networks in Fig. 7.

Protein	Best blast Ath	Centrali-ty t10-1	Rewir-ing t10-1	Score	Centrali-ty t11	Rewir-ing t11	Score
Glyma.10G173000	TOI5	1.265	62.5	2	0.599	35	2
Glyma.19G007300	TOT3	1.029	34.5	1.29	0.201	14	0.79
Glyma.08G037300	CDKF;1	0.259	15.5	0.33	0.218	6.5	0.51
Glyma.13G161700	CRCK3	0.167	44	0.79	0.206	21.5	0.94
Glyma.06G161200	CPK4	0.057	13	0.12	0	7.5	0.18
Glyma.15G074900	AT5G58950	0.007	24	0.29	0.004	9	0.23
Glyma.20G070700	CPK1	0.002	16	0.13	0.006	8	0.20
Glyma.07G046800	SRF6	0	9	0	0	1.5	0
Glyma.07G072100	BSK3	0	13.5	0.08	0	5.5	0.12

We included this comparison in the methods; Line 779 to 785: “Given our time course (every 6 minutes for one hour) and the fast nature of phosphorylation cascades, we selected time lapse 0 and 1 as we assumed that the regulation may happen within and between time points. To substantiate the choice of including both time lapse 0 and 1, we also proceeded with only time lapse 1, compared network topology, and were able to draw the same conclusions

(Supplemental Fig. S8). For time course datasets with smaller time steps, only the inclusion of time lapse 1 is advised (Spurney et al., 2021).”.

To note, we kept the possibility to include time lapse 0, which assumes a phosphorylation within the same time point, and a time lapse 1, which assumes a phosphorylation at the next time point, in our R package. Users can thus make critical decision on which time lapse suits their datasets the best.

Fig. 7 (Included in the manuscript as Supplemental Fig. S8) – Phosphorylation cascades upon cold in soybean. (A) Signaling network that includes regulation inferred based on both the simultaneous and antecedent phosphorylation change of the kinases/phosphatases when compared with the phosphorylation change of their potential targets. (B) Signaling network that only includes the regulation based on the antecedent phosphorylation change of the kinases/phosphatases when compared with the phosphorylation change of their potential targets. Orange, blue, and grey nodes represent phosphosites present in the control, cold, or both networks, respectively. Round and triangle nodes represent kinases/phosphatases and other phosphosites, respectively.

Minor comment

Line 150 biLSTM acronym not defined. Please write out in full

REPLY: We spelled out biLSTM at first use in the first result section: “The neural network architecture consists of four different layers: (1) a convolutional neural network layer (CNN) that extracts putative protein domains, i.e. the functional units of a protein, by performing a convolution across the sequences with a kernel size of 7 (Fig. 1A, see Methods), (2) an attention layer that emphasizes the learned patterns from the CNN layer by assigning increased importance to key domains, (3) a bidirectional long short term memory (biLSTM) layer that captures long-distance dependencies within the sequences and between detected domains, and (4) two dense layers connected to the output vector (Fig. 1A).”

Line 724 formula lost it’s parameters

REPLY: Thank you for pointing out this discrepancy with the variables. An error occurred when uploading the PDF to the submission platform. We have corrected the issue.

References

Lopez-del Rio, A., Martin, M., Perera-Lluna, A. et al. Effect of sequence padding on the performance of deep learning models in archaeal protein functional prediction. *Sci Rep* 10, 14634 (2020) doi: <https://doi.org/10.1038/s41598-020-71450-8>

Koonin V. Eugene. Orthologs, Paralogs, and Evolutionary Genomics. *Annual Review of Genetics* 2005 39:1, 309-338 doi: 10.1146/annurev.genet.39.073003.114725

V. Satopaa, J. Albrecht, D. Irwin and B. Raghavan, "Finding a "Kneedle" in a Haystack: Detecting Knee Points in System Behavior," 2011 31st International Conference on Distributed Computing Systems Workshops, Minneapolis, MN, USA, 2011, pp. 166-171, doi: 10.1109/ICDCSW.2011.20.

Nadav Brandes, Dan Ofer, Yam Peleg, Nadav Rappoport, Michal Linial, ProteinBERT: a universal deep-learning model of protein sequence and function, *Bioinformatics*, Volume 38, Issue 8, March 2022, Pages 2102–2110, <https://doi.org/10.1093/bioinformatics/btac020>

Sievers F, Wilm A, Dineen D, et al. Fast, scalable generation of high-quality protein multiple sequence alignments using Clustal Omega. *Molecular Systems Biology*. 2011 Oct;7:539. DOI: 10.1038/msb.2011.75. PMID: 21988835; PMCID: PMC3261699.

Tyanova, S., Temu, T., Sinitcyn, P. et al. The Perseus computational platform for comprehensive analysis of (prote)omics data. *Nat Methods* 13, 731–740 (2016). <https://doi.org/10.1038/nmeth.3901>

Karpievitch, Y.V., Dabney, A.R. & Smith, R.D. Normalization and missing value imputation for label-free LC-MS analysis. *BMC Bioinformatics* 13 (Suppl 16), S5 (2012). <https://doi.org/10.1186/1471-2105-13-S16-S5>

Vu LD, Xu X, Zhu T, Pan L, van Zanten M, de Jong D, Wang Y, Vanremoortele T, Locke AM, van de Cotte B, De Winne N, Stes E, Russinova E, De Jaeger G, Van Damme D, Uauy C, Gevaert K, De Smet I. The membrane-localized protein kinase MAP4K4/TOT3 regulates thermomorphogenesis. *Nat Commun*. 2021 May 14;12(1):2842. doi: 10.1038/s41467-021-23112-0. PMID: 33990595; PMCID: PMC8121802.

Nikonorova N, Van den Broeck L, Zhu S, van de Cotte B, Dubois M, Gevaert K, Inzé D, De Smet I. Early mannitol-triggered changes in the Arabidopsis leaf (phospho)proteome reveal growth regulators. *J Exp Bot*. 2018 Aug 31;69(19):4591-4607. doi: 10.1093/jxb/ery261. PMID: 30010984; PMCID: PMC6117580.

Smith S, Zhu S, Joos L, Roberts I, Nikonorova N, Vu LD, Stes E, Cho H, Larrieu A, Xuan W, Goodall B, van de Cotte B, Waite JM, Rigal A, Ramans Harborough S, Persiau G, Vanneste S, Kirschner GK, Vandermarliere E, Martens L, Stahl Y, Audenaert D, Friml J, Felix G, Simon R, Bennett MJ, Bishopp A, De Jaeger G, Ljung K, Kepinski S, Robert S, Nemhauser J, Hwang I, Gevaert K, Beeckman T, De Smet I. The CEP5 Peptide Promotes Abiotic Stress Tolerance, As Revealed by Quantitative Proteomics, and Attenuates the AUX/IAA Equilibrium in Arabidopsis. *Mol Cell Proteomics*. 2020 Aug;19(8):1248-1262. doi: 10.1074/mcp.RA119.001826. Epub 2020 May 13. PMID: 32404488; PMCID: PMC8011570.

Nikonorova N, Murphy E, Fonseca de Lima CF, Zhu S, van de Cotte B, Vu LD, Balcerowicz D, Li L, Kong X, De Rop G, Beeckman T, Friml J, Vissenberg K, Morris PC, Ding Z, De Smet I. The

Arabidopsis Root Tip (Phospho)Proteomes at Growth-Promoting versus Growth-Repressing Conditions Reveal Novel Root Growth Regulators. *Cells*. 2021 Jul 2;10(7):1665. doi: 10.3390/cells10071665. PMID: 34359847; PMCID: PMC8303113.

Min Zou, Suzanne D. Conzen, A new dynamic Bayesian network (DBN) approach for identifying gene regulatory networks from time course microarray data, *Bioinformatics*, Volume 21, Issue 1, January 2005, Pages 71–79, <https://doi.org/10.1093/bioinformatics/bth463>

Blazek M, Santisteban TS, Zengerle R, Meier M. Analysis of fast protein phosphorylation kinetics in single cells on a microfluidic chip. *Lab Chip*. 2015 Feb 7;15(3):726-34. doi: 10.1039/c4lc00797b. PMID: 25428717.

Sullivan, S., Waksman, T., Paliogianni, D. et al. Regulation of plant phototropic growth by NPH3/RPT2-like substrate phosphorylation and 14-3-3 binding. *Nat Commun* 12, 6129 (2021). <https://doi.org/10.1038/s41467-021-26333-5>

Schulze, Birgit et al. Rapid Heteromerization and Phosphorylation of Ligand-activated Plant Transmembrane Receptors and Their Associated Kinase BAK1. 2010 *Journal of Biological Chemistry*, Volume 285, Issue 13, 9444 – 9451

Smith JM, Salamango DJ, Leslie ME, Collins CA, Heese A. Sensitivity to Flg22 is modulated by ligand-induced degradation and de novo synthesis of the endogenous flagellin-receptor FLAGELLIN-SENSING2. *Plant Physiol*. 2014 Jan;164(1):440-54. doi: 10.1104/pp.113.229179. Epub 2013 Nov 12.

Cargnello M, Roux PP. Activation and function of the MAPKs and their substrates, the MAPK-activated protein kinases. *Microbiol Mol Biol Rev*. 2011 Mar;75(1):50-83. doi: 10.1128/MMBR.00031-10. Erratum in: *Microbiol Mol Biol Rev*. 2012 Jun;76(2):496. PMID: 21372320; PMCID: PMC3063353.

Picton C, Aitken A, Bilham T, Cohen P. Multisite phosphorylation of glycogen synthase from rabbit skeletal muscle. Organisation of the seven sites in the polypeptide chain. *Eur J Biochem*. 1982 May;124(1):37-45. doi: 10.1111/j.1432-1033.1982.tb05903.x. PMID: 6806097.

Spurney, R., Schwartz, M., Gobble, M., Sozzani, R., Van den Broeck, L. (2021). Spatiotemporal Gene Expression Profiling and Network Inference: A Roadmap for Analysis, Visualization, and Key Gene Identification. In: MUKHTAR, S. (eds) *Modeling Transcriptional Regulation*. *Methods in Molecular Biology*, vol 2328. Humana, New York, NY. https://doi.org/10.1007/978-1-0716-1534-8_4

Huynh-Thu VA, Irrthum A, Wehenkel L, Geurts P (2010) Inferring Regulatory Networks from Expression Data Using Tree-Based Methods. *PLOS ONE* 5(9): e12776. <https://doi.org/10.1371/journal.pone.0012776>

Park, S., Koh, Y., Jeon, H. et al. Enhancing the interpretability of transcription factor binding site prediction using attention mechanism. *Sci Rep* 10, 13413 (2020). <https://doi.org/10.1038/s41598-020-70218-4>

Storey, J. D., Bass, A. J., Dabney, A. & Robinson, D. qvalue: Q-value estimation for false discovery rate control. (R, 2021).

REVIEWERS' COMMENTS

Reviewer #1 (Remarks to the Author):

The author has revised the manuscript according to the comments. It could be published now.

Reviewer #3 (Remarks to the Author):

I've read the response to the reviewers and revised manuscript. The authors have carefully and thoroughly addressed all questions and comments and I have no further concerns. I commend the authors for their thoughtful approach and elaborate response to all comments.